# PCEval: A Benchmark for Evaluating Physical Computing Capabilities of Large Language Models

## Abstract

Large Language Models (LLMs) have demonstrated remarkable capabilities across various domains, including software development, education, and technical assistance. Among these, software development is one of the key areas where LLMs are increasingly adopted. However, when hardware constraints are considered—for instance, in physical computing, where software must interact with and control physical hardware —their effectiveness has not been fully explored. To address this gap, we introduce PCEval (Physical Computing Evaluation), the first benchmark in physical computing that enables a fully automatic evaluation of the capabilities of LLM in both the logical and physical aspects of the projects, without requiring human assessment. Our evaluation framework assesses LLMs in generating circuits and producing compatible code across varying levels of project complexity. Through comprehensive testing of 13 leading models, PCEval provides the first reproducible and automatically validated empirical assessment of LLMs' ability to reason about fundamental hardware implementation constraints within a simulation environment. Our findings reveal that while LLMs perform well in code generation and logical circuit design, they struggle significantly with physical breadboard layout creation, particularly in managing proper pin connections and avoiding circuit errors. PCEval advances our understanding of AI assistance in hardware-dependent computing environments and establishes a foundation for developing more effective tools to support physical computing education.

## 1 Introduction

Physical computing is the practice of connecting software with the physical world, typically through micro-controller platforms such as Arduino that control sensors, actuators, and displays. This connection reveals how implemented code produces tangible effects in the physical world, effectively bridging abstract computation and real-world objects. This unique characteristic has made physical computing a rapidly growing element of modern STEM (Science, Technology, Engineering, and Mathematics) education, valued for fostering creativity, problem-solving skills, and computational thinking through hands-on experiences (Chung & Lou, 2021; Kastner-Hauler et al., 2022; Araújo & Saúde, 2025). Consequently, educational institutions worldwide are increasingly offering physical computing classes to help students interact with technology and understand complex concepts (El-Abd, 2017; García-Tudela & Marín-Marín, 2023; Schätz et al., 2024).

Despite its educational value, physical computing remains difficult to teach and assess. In physical computing education, teachers must have dual expertise in software and hardware, yet even experienced educators report challenges in evaluating physical circuits with their small components and tangled wires (Hyeon et al., 2016; Theodoropoulos et al., 2018). In addition, our interviews with eight experienced computer science educators (Appendix A) showed that individualized feedback and circuit verification represent a widespread burden in classroom practice.

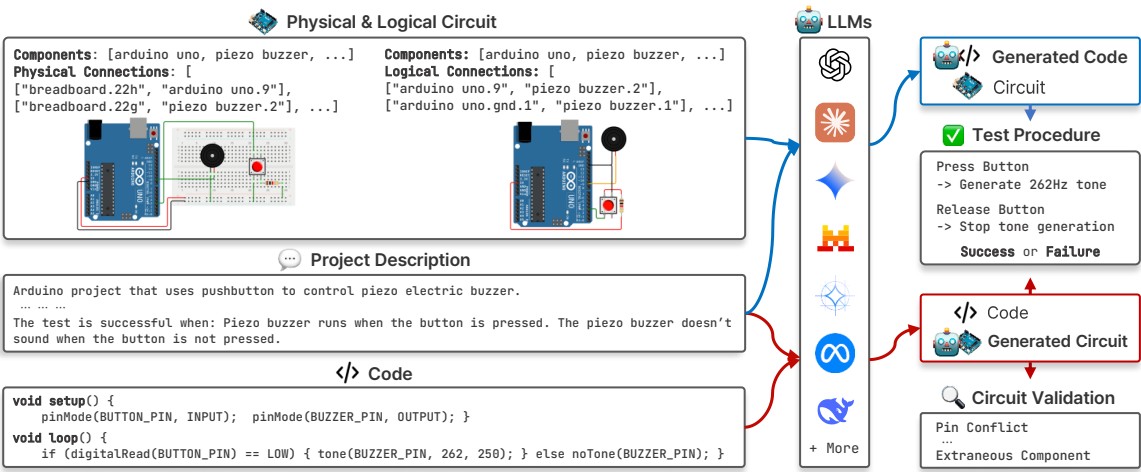

Figure 1: This figure illustrates the core protocols of the PCEVAL benchmark. The left panels show the input options for LLMs, while the right panels show the evaluation workflow. There are basically two types of tasks: code generation (blue arrow) and circuit generation (red arrow). Generated outputs undergo validation through standardized test procedures and circuit validation metrics, measuring both logical correctness and physical implementation feasibility in Arduino systems. (The circuit visualizations are shown to aid understanding; they are not used as inputs to the LLMs.)

Given these persistent challenges, it is natural to ask whether recent advances in AI, such as Large Language Models (LLMs), could provide a solution. However, although LLMs have demonstrated remarkable capabilities in software development and coding assistance (Annepaka & Pakray, 2024; Wang et al., 2024), their effectiveness in hardware-dependent environments, such as physical computing, has not been fully explored. Prior efforts on LLMs for electronics (Jansen, 2023; Yang et al., 2024) further highlight this gap, as they relied on labor-intensive manual expert evaluation and largely overlooked the physical layout constraints central to educational breadboard implementations.

*"In physical computing, working with circuits is absolutely essential, and that's just something AI can't help with. These days, tools like Gemini in Colab can write and even suggest code, which is great. But it's not like an arm can suddenly pop out and build the circuit for you."*

- Public High School Computer Science Teacher C, Appendix A. Q4

To address this gap, we introduce PCEVAL (Physical Computing Evaluation), the first benchmark designed to evaluate LLMs' abilities in both the logical and physical aspects of physical computing tasks. PCEVAL decomposes the challenge into four dimensions: (1) Logical Circuit Generation, (2) Physical Circuit Generation, (3) Code from Logical Circuits, and (4) Code from Physical Circuits. This structured decomposition enables **scalable and reproducible evaluation without manual expert intervention**, overcoming the subjectivity and labor costs of prior manual assessments.

Our evaluation of leading LLM architectures on PCEVAL spans projects of varying complexity, from basic I/O operations to intricate multi-component interactive systems. The experimental results reveal a clear disparity: while LLMs demonstrate competence in code generation and logical circuit design, they struggle considerably with physical circuit generation, particularly in adhering to breadboard mechanics and ensuring correct pin configurations. Although mitigation strategies such as self-improvement and chain-of-thought prompting provided partial improvements, substantial challenges remain, reflecting the difficulty of physical computing

tasks. By illuminating these challenges, PCEVAL establishes a critical starting point for developing models capable of truly interacting with our physical world.

This research offers several key contributions:

- **New Benchmark (PCEVAL):** We introduce the first comprehensive benchmark for physical computing that systematically evaluates both logical reasoning and physical implementation capabilities.
- **Scalable Automated Evaluation:** Our framework eliminates subjective manual assessment through decomposed tasks and standardized metrics, enabling reproducible evaluation at scale.
- **Physical Gap Identification:** We reveal limitations in current LLMs' physical constraint reasoning, providing actionable insights for developing more effective AI tools for hardware-dependent domains.

## 2 RELATED WORK

**AI for Programming.** Early LLM benchmarks focused on algorithmic tasks in Python, later extended to broader domains like data science and web/mobile applications (Chen et al., 2021; Austin et al., 2021; Hendrycks et al., 2021; Lai et al., 2023; Zheng et al., 2024). While these works advanced functional correctness in software, they do not address the challenges of hardware interaction or physical implementation.

**AI for Hardware and Physical Systems.** Recent research has explored LLMs in hardware-related domains, including Verilog/HDL generation (Liu et al., 2023; Thakur et al., 2024), analog circuit design with SPICE-based evaluation (Lai et al., 2025), and PCB layout assistance (Liu et al., 2025). Educational contexts such as Arduino programming have also been considered (Johnson et al., 2024; Subramanium et al., 2024), but these studies typically limit themselves to small-scale code snippets without evaluating physical circuit design. EmbedGenius (Yang et al., 2024) introduced hardware-in-the-loop testing for IoT systems, yet its scope is restricted to validating code against pre-defined circuit representations, bypassing the problem of generating physically valid circuits. MICRO25 (Jansen, 2023) is closest to our work, demonstrating that LLMs can generate electronic device designs from text. However, its evaluation depends heavily on *manual expert judgment*, raising concerns about cost, scalability, and reproducibility. Furthermore, MICRO25 does not address physical circuit generation and assess code generation under physical constraints. In contrast, PCEVAL introduces two key advances: (1) a fully automated and reproducible evaluation pipeline ensuring scalability and consistency; and (2) a comprehensive focus on physical computing, uniquely covering both logical/physical circuit generation and code generation from physical layouts. PCEVAL addresses both gaps through automated evaluation with decomposed tasks and standardized metrics, while uniquely assessing physical circuit generation and code generation from physical layouts—capabilities essential for physical computing support. Thus, we believe PCEVAL is not just another benchmark, but the first tool for holistic assessment of LLMs in physical computing, enabling scalable progress in the field.

## 3 PCEVAL BENCHMARK

### 3.1 THE INTERVIEWS AND PROBLEM DEFINITION

To develop a benchmark addressing real challenges in physical computing education, we first conducted in-depth interviews with eight experienced CS educators (Appendix A). These discussions revealed several obstacles in physical computing education. Educators consistently highlighted three critical challenges: (1) time-intensive hardware setup and debugging (A, B, C, E), (2) difficulties providing individualized support across varying student skill levels (A, C, D, E, F, G, H), and (3) the complex interplay between circuit construction and code functionality (A, B, C). For instance, public school CS Teacher A candidly noted the inherent manageability constraints, stating, "*I think the maximum number of students one teacher can*

Table 1: Comparative analysis of related benchmarks, highlighting key differentiators in task scope and evaluation. *For VerilogEval and ResBench, code generation itself constitutes the logical circuit description (HDL). [†]EmbedTask's automated evaluation via HIL testing requires physical hardware setup. [§]PCEVAL's code generation assessment uniquely includes tasks for generating code from detailed physical circuit layouts.

| Benchmark | Target Domain | Code Generation | Logical Circuit Generation | Physical Circuit Generation | Automated Evaluation | Validation Method |
|---|---|---|---|---|---|---|
| HumanEval (Chen et al., 2021) | General SW | ✓ | ✗ | ✗ | ✓ | Unit Testing |
| MBPP (Austin et al., 2021) | | ✓ | ✗ | ✗ | ✓ | Unit Testing |
| VerilogEval (Liu et al., 2023) | HDL Design | ✓ | ✓* | ✗ | ✓ | HDL Simulation |
| ResBench (Guo & Zhao, 2025) | | ✓ | ✓* | ✗ | ✓ | HDL Sim. & Synthesis |
| EmbedTask (Yang et al., 2024) | | ✓ | ✗ | ✗ | ✓[†] | HIL Testing |
| MICRO25 (Jansen, 2023) | Physical Computing | ✓ | ✓ | ✗ | ✗ | Manual Evaluation |
| **PCEVAL** | | ✓[§] | ✓ | ✓ | ✓ | HW/SW Sim. |

*handle in physical computing class is about 15.*" These firsthand accounts indicate that while physical computing offers immense educational potential (Appendix A.Q4), its practical implementation is fraught with significant, persistent hurdles. Addressing these hurdles effectively may require new forms of assistance, potentially from AI, but this first necessitates a clear understanding of current AI capabilities and limitations within this specific context.

In particular, the interviews reveal that practical challenges in physical computing directly shape the technical problems of PCEVAL, the benchmark designed to evaluate LLMs in this domain. While current LLMs are adept at software tasks (Chen et al., 2021), they struggle to integrate abstract instructions with tangible, constrained hardware in physical computing. Key deficiencies include reasoning about physical constraints (e.g., layouts, pinouts) and ensuring coherence between circuit designs and corresponding code. Accordingly, PCEVAL systematically assesses LLMs not only on isolated code or schematic generation but, crucially, on producing physically viable layouts and hardware-compatible code, providing guidance for AI tool development in this domain.

## 3.2 DATASET STRUCTURE

Each PCEVAL project instance is defined by five components: **(D)escription**, a natural language specification of objectives and requirements; **(L)ogical circuit**, an abstract pin-to-pin connection map; **(P)hysical circuit**, a breadboard-level implementation with wiring constraints; **(C)ode**, an executable program aligned with the given circuit; and **(T)est procedure**, a set of automated checks that assert expected outputs under simulated inputs. This decomposition enables controlled evaluation of specific LLM capabilities while maintaining realistic educational scenarios.

## 3.3 TASK DEFINITIONS

The PCEVAL benchmark is designed to rigorously assess the capabilities of LLMs in physical computing through four distinct generation tasks. As illustrated in Figure 1, each task challenges an LLM to produce a specific target artifact (either a circuit representation or code) based on a designated set of input components from our dataset structure. These tasks systematically probe different facets of an LLM's understanding, from logical design to physical implementation and code-hardware compatibility. The four tasks are defined as follows:

1. **Logical Circuit Generation** ($D, C \rightarrow L$): Given a natural language project description $D$ and the corresponding code $C$, the LLM is tasked with generating a complete logical circuit specification $L$. This task primarily evaluates the LLM's ability to infer necessary hardware components and their abstract,

pin-to-pin logical connections based on functional requirements and program logic, without concern for physical layout details.

2. **Physical Circuit Generation** ($D, C \rightarrow P$): Using the same inputs as the previous task, the project description $D$ and code $C$, the LLM must generate a detailed physical circuit layout $P$ suitable for breadboard implementation. This task assesses the LLM's understanding of practical hardware integration, including valid wire routing adhering to breadboard mechanics and physical constraints.

3. **Code Generation from Logical Circuit** ($D, L \rightarrow C$): In this task, the LLM is provided with the project description $D$ and a logical circuit specification $L$. Its objective is to generate a functional code $C$ that correctly implements the project requirements using the provided logical hardware structure. This tests the ability to translate a conceptual circuit design into working code.

4. **Code Generation from Physical Circuit** ($D, P \rightarrow C$): Similar to the preceding task, the LLM receives the project description $D$, but in this case, it is accompanied by a physical circuit layout $P$. The LLM must generate the code $C$ that is compatible with this specific physical hardware configuration, including adherence to the explicit pin assignments dictated by the breadboard layout.

For each project in PCEVAL, we evaluate all four tasks using a controlled methodology that isolates the specific generation capability being assessed. As shown in the right panel of Figure 1, we pair each type of LLM-generated artifact with one of the reference components ($L$, $P$, or $C$) to construct complete, testable systems. When evaluating circuit generation capabilities, we connect the generated circuit with the reference code. For code generation tasks, we pair the generated code with the corresponding reference circuit. This structured evaluation approach allows us to execute the Test Procedure ($T$) within the simulation environment (Shaked, 2020) and precisely measure performance on each distinct capability. By systematically controlling one aspect while testing another, we can accurately determine whether the integrated system functions correctly according to project requirements, ensuring a focused assessment of the LLM's performance across the physical computing spectrum.

### 3.4 PROJECT DESIGN

The PCEVAL benchmark comprises 50 projects designed to reflect authentic educational scenarios in physical computing, covering commonly utilized components such as LEDs, sensors, and displays. To facilitate systematic evaluation of LLM capabilities, we categorize projects into four complexity levels (Table 2), ranging from single-component control (Level 1) to multi-component system design (Level 4). This structure ensures that the dataset not only spans a broad technical spectrum but also aligns with typical physical computing curricula in middle and high school settings, where teachers often cover only two to three projects per semester.

Table 2: Summary statistics for the PCEval benchmark dataset, categorized by complexity level, showing project counts, code and circuit complexity metrics.

| Statistic | Level 1 | Level 2 | Level 3 | Level 4 |
|---|---|---|---|---|
| Num. Projects | 12 | 13 | 14 | 11 |
| ($C$) Lines of Code | 14.00 | 18.92 | 19.89 | 27.00 |
| ($C$) Cyclomatic Complexity | 3.58 | 5.92 | 5.93 | 7.18 |
| ($L$) Num. Components | 3.83 | 3.77 | 6.43 | 8.18 |
| ($L$) Num. Connections | 7.25 | 7.00 | 15.64 | 16.36 |
| ($P$) Num. Components | 4.83 | 4.77 | 7.43 | 9.18 |
| ($P$) Num. Connections | 15.42 | 15.85 | 35.86 | 35.0 |

By encompassing representative tasks across all levels of difficulty, the 50 projects provide sufficient breadth and depth to approximate the scope of a full semester course, while remaining tractable for reproducible evaluation. Detailed project descriptions and examples are provided in Appendix E, along with additional discussion of dataset scale.

Table 3: Comparative performance (success rate %) of various LLMs on PCEVAL benchmark tasks, high-lighting differential capabilities across circuit generation and code generation domains.

| Model | Param. | Circuit Gen. | | | Code Gen. | | | Total Overall |
|---|---|---|---|---|---|---|---|---|
| | | $D, C \rightarrow L$ | $D, C \rightarrow P$ | Overall | $D, L \rightarrow C$ | $D, P \rightarrow C$ | Overall | |
| *Closed Source LLMs* | | | | | | | | |
| GPT-4o-mini | - | 48.0 | 1.2 | 24.6 | 49.2 | 51.2 | 50.2 | 37.4 |
| Claude 3.5 Haiku | - | 47.6 | 1.6 | 24.6 | 60.0 | 55.6 | 57.8 | 41.2 |
| Gemini-2.0-Flash-Lite | - | 50.0 | 2.4 | 26.2 | 54.8 | 56.4 | 55.6 | 40.9 |
| Gemini-2.0-Flash | - | 58.4 | 19.6 | 39.0 | 54.4 | 50.4 | 52.4 | 45.7 |
| GPT-4.1 | - | 50.4 | 12.0 | 31.2 | 65.2 | 63.2 | 64.2 | 47.7 |
| GPT-4o | - | 58.0 | 26.8 | 42.4 | 61.2 | 56.4 | 58.8 | 50.6 |
| Claude 3.7 Sonnet | - | 65.6 | 13.6 | 39.6 | 62.8 | 64.0 | 63.4 | 51.5 |
| o3-mini | - | **66.0** | **45.2** | **55.6** | **67.6** | **68.0** | **67.8** | **61.7** |
| *Open Source LLMs* | | | | | | | | |
| LLaMA 3.1 | 8B | 21.6 | 2.0 | 11.8 | 25.6 | 24.0 | 24.8 | 18.3 |
| DeepSeek-Coder-V2 | 16B | 26.0 | 1.2 | 13.6 | 30.0 | 20.8 | 25.4 | 19.5 |
| Gemma 3 | 27B | 45.2 | 2.4 | 23.8 | 32.4 | 28.4 | 30.4 | 27.1 |
| Phi 4 | 14B | 30.0 | 2.8 | 16.4 | 44.8 | **35.6** | 20.1 | 28.3 |
| Mistral-Small 3 | 24B | **46.4** | **13.6** | **30.0** | 45.6 | 30.8 | **38.2** | **34.1** |

## 3.5 EVALUATION

We assess LLM performance across all PCEVAL tasks using a multi-faceted approach that prioritizes both functional correctness and implementation feasibility. Our primary evaluation metric for all tasks is simulation success, which verifies whether the generated artifacts operate correctly according to the project specifications by executing test procedures within a simulation environment (Shaked, 2020). Our circuit validation protocol identifies errors (redundant connections, extraneous/missing components, isolated components) in logical and physical circuit designs. For physical circuit generation specifically, success requires both passing the simulation test procedure and avoiding implementation errors (pin conflicts and breadboard bypasses) that would make physical construction impossible despite simulation success. To ensure statistical reliability, we conduct five independent trials for each task-model combination and report averaged results. We provide detailed evaluation metrics in Appendix C.

## 4 EXPERIMENTAL RESULTS

### 4.1 SETUP

We evaluated a diverse set of closed- and open-source LLMs, including GPT-4o, Claude 3.7 Sonnet, o3-mini, Gemini-2.0-Flash, as well as leading open-source models such as LLaMA 3.1, Mistral-Small, and Phi 4 (Team et al., 2025; Grattafiori et al., 2024; Zhu et al., 2024; Abdin et al., 2024; Jiang et al., 2023). For each of the four benchmark tasks, we used standardized prompts with explicit output specifications (Appendix F).

### 4.2 ANALYSIS OF MODEL PERFORMANCE AND TASK CHARACTERISTICS

**Circuit Gen. vs. Code Gen.** Table 3 shows that most models achieve higher success in *code* tasks than in *circuit* tasks. For example, Claude 3.7 Sonnet reaches 63.4% on code overall vs. 39.6% on circuit overall; GPT-4.1 reaches 64.2% vs. 31.2%. Even the strongest model (o3-mini) records 45.2% on physical-circuit generation ($D,C \rightarrow P$) vs. 66.0% on logical-circuit generation ($D,C \rightarrow L$). Overall, generating physically valid layouts remains substantially harder than generating code.

Table 4: Detailed success rates (%) of selected high-performing LLMs across project complexity levels, demonstrating performance degradation with increasing project complexity.

| Model | $D, C \rightarrow L$ | | | | $D, C \rightarrow P$ | | | | $D, L \rightarrow C$ | | | | $D, P \rightarrow C$ | | | |
|---|---|---|---|---|---|---|---|---|---|---|---|---|---|---|---|---|
| | L1 | L2 | L3 | L4 | L1 | L2 | L3 | L4 | L1 | L2 | L3 | L4 | L1 | L2 | L3 | L4 |
| Gemini-2.0-Flash | 81.7 | 61.5 | 35.7 | 58.2 | 36.7 | 33.8 | 1.4 | 7.3 | 71.7 | 63.1 | 45.7 | 36.4 | 61.7 | 53.8 | 45.7 | 40.0 |
| Claude 3.7 Sonnet | **86.7** | 63.1 | **58.6** | 54.5 | 33.3 | 15.4 | 1.4 | 5.5 | **81.7** | 61.5 | **64.3** | 41.8 | **83.3** | 56.9 | **71.4** | 41.8 |
| GPT-4o | 78.3 | 55.4 | 37.1 | **65.5** | 56.7 | 30.8 | 8.6 | 12.7 | 76.7 | 63.1 | 50.0 | 56.4 | 75.0 | 63.1 | 44.3 | **43.6** |
| o3-mini | 85.0 | **69.2** | 54.3 | 56.4 | **65.0** | **60.0** | **28.6** | **25.5** | 78.3 | **70.8** | 61.4 | **60.0** | 76.7 | **70.8** | 64.3 | 25.5 |
| Mistral-Small 3 | 66.7 | 50.8 | 35.7 | 32.7 | 36.7 | 15.4 | 0.0 | 3.6 | 56.7 | 56.9 | 28.6 | 41.8 | 50.0 | 43.1 | 20.0 | 9.1 |

**Logical Circuit Gen. and Physical Circuit Gen.** The most striking finding, evident in Table 3, is the substantial performance gap between logical and physical circuit generation. Across all models, success rates for Physical Circuit Generation ($D, C \rightarrow P$) are markedly lower than for Logical Circuit Generation ($D, C \rightarrow L$). Many models, including several prominent ones, achieved success rates below 10% for physical circuit generation. Even top performers like o3-mini (45.2%) and Claude 3.7 Sonnet (13.6%) found this task considerably more challenging than logical circuit design (where they scored 66.0% and 65.6%, respectively). This highlights the profound difficulty LLMs face in translating conceptual project requirements and code into physically valid and implementable breadboard layouts while considering pin assignments.

**Sensitivity to Complexity.** The difficulty compounds with project complexity (Table 4). For physical circuit generation, success commonly drops from Level 1 (L1) to Level 4 (L4): o3-mini 65.0% → 25.5%, GPT-4o 56.7% → 12.7%, Claude 3.7 33.3% → 5.5%, Mistral-Small 3 36.7% → 3.6%. Logical circuit and code tasks also degrade with level, but far less precipitously. This aligns with the dataset's complexity gradient (Table 2): higher levels introduce more components and denser interconnections, stressing spatial reasoning and pin-allocation consistency.

**Dominant Failure Causes in $D, C \rightarrow P$.** Physical circuit generation errors are not merely logical inconsistencies; they reflect violations of *breadboard mechanics*. Figure 2 summarizes two primary constraints—pin conflicts and breadboard bypasses—that most strongly depress success. Average pin-conflict counts per sample are highest among all error types (e.g., Claude 3.7: 7.52, o3-mini: 4.20, GPT-4o: 2.07), while bypasses vary by model (e.g., Gemini-2.0-Flash: 2.73 vs. o3-mini: 0.01; see de-

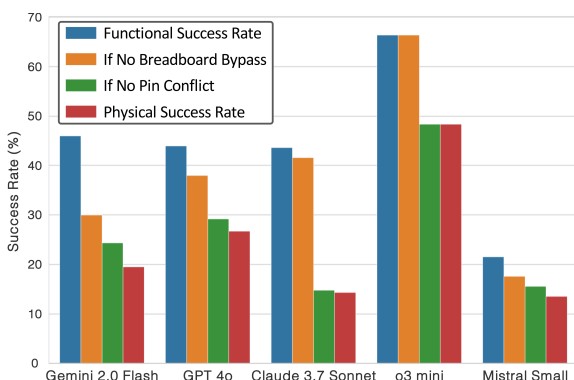

Figure 2: Impact of physical-constraint filters in $D$, $C \rightarrow P$: success rates are shown under progressively stricter criteria— **blue**: functional correctness only; **orange**: functional correctness *and* no breadboard bypass; **green**: functional correctness *and* no pin conflict; **red**: functional correctness *and* no breadboard bypass *and* no pin conflict. The drop from blue to red highlights how bypasses and pin conflicts drive failures in physical layout generation.

tails in Appendix C. Other integrity issues (extraneous/isolated/missing components) occur in both logical and physical circuit tasks, but their magnitudes are smaller and less predictive of outright failure than pin conflicts. **Implication:** the core weakness is *physical-constraint reasoning*, especially allocating/routeing connections without violating shared nodes and board topologies.

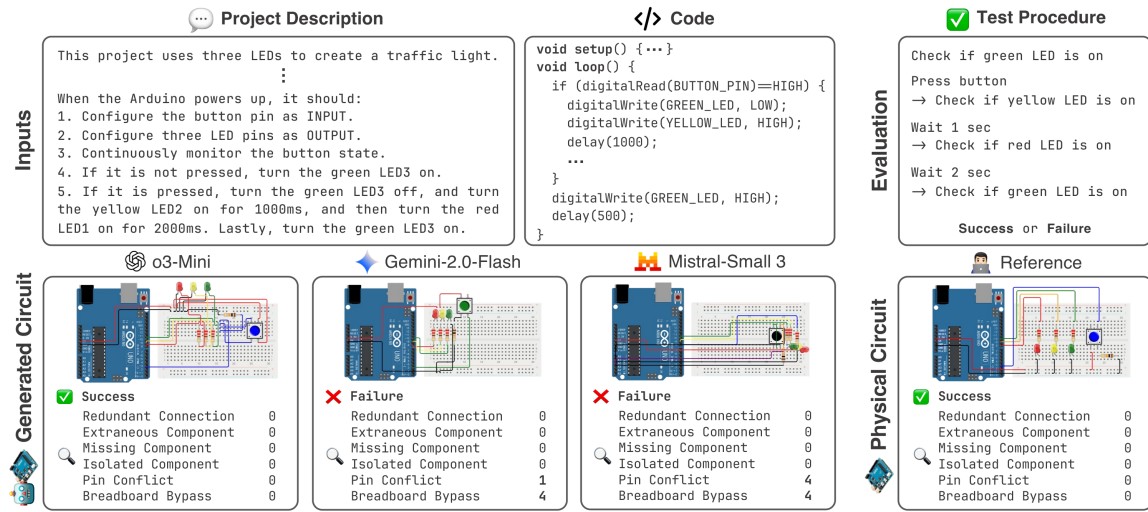

Figure 3: Qualitative examples of LLM-generated physical circuits for a traffic light project (level 4), illustrating successful and failed attempts with corresponding error analyses.

## 4.3 EXPLORATORY MITIGATION STRATEGIES

**Self-improvement.** Our error analysis revealed that many LLM failures stemmed from surprisingly simple issues such as missing connections and incorrect pin number. Motivated by this, we explored a multi-turn self-improvement strategy. When the initial output failed, the LLM received structured failure logs and iteratively refined its solution (up to five turns) to correct such minor errors. Applied across all four tasks, this approach yielded consistent gains, (e.g., o3-mini improved from 61.7% to 76.5%; see Appendix D).

**Chain-of-thought prompting.** For tasks which use physical circuit, we hypothesized that explicit intermediate reasoning could help bridge logical design and physical implementation. We therefore guided models to first generate an intermediate Logical Circuit representation before producing the final code or physical layout—mirroring the human process of reasoning from abstract connections to concrete breadboard placement. This chain-of-thought strategy produced notable improvements for some models (e.g., GPT-4o: +10.4%, Mistral-Small 3: +18.0% on physical code task), highlighting its promise as a complementary technique, though its impact was not uniform across all models (Appendix D).

## 5 EDUCATIONAL USABILITY AND PEDAGOGICAL IMPLICATIONS

Our findings suggest that LLMs hold considerable promise for physical computing education, but their integration must be guided by careful pedagogical considerations. Prior studies already caution that students may rely on LLMs for "ready-made answers," which can hinder deeper engagement and learning (Anson, 2024; Jošt et al., 2024). This concern was also raised in our *initial interviews with eight experienced CS teachers*, which we conducted at the outset to identify the central pain points in physical computing education (Appendix A). Educators emphasized that while LLMs can generate working Arduino solutions, students may bypass essential reasoning processes if guidance is not structured. This challenge is amplified by known factors such as automation bias (Li & Wu, 2025) and the moderating role of prior knowledge (Lee et al., 2025), where learners with limited foundational skills may be less able to critically assess AI outputs.

To complement these design-stage insights, we conducted *post-benchmark validation studies* after developing PCEVAL. Specifically, we organized a small focus group with two physical computing educators (8 and 10 years of experience, 30 minutes each) and a survey with three pre-service CS teachers (Appendix B). Unlike the initial interviews, these sessions were conducted *with the benchmark and model outputs in hand*, enabling participants to directly assess educational potential and limitations. Both groups highlighted tangible benefits—particularly the use of automated circuit verification to reduce instructor workload and assist less-experienced teachers. At the same time, they noted practical barriers for classroom use: the lack of step-by-step assembly guidance, declining readability in complex layouts, and the absence of pedagogical conventions such as consistent wire colors. Notably, pre-service teachers rated low-complexity projects as highly usable in classrooms, but their educational value dropped sharply at higher complexity levels.

Taken together, these findings suggest a dual imperative. First, LLMs should be positioned not merely as answer generators but as scaffolding tools that structure learning, encourage exploration, and maintain student agency in problem-solving. Second, benchmarks like PCEVAL should evolve beyond functional correctness to incorporate educational usability metrics—including readability conventions, step-by-step guidance, and pedagogical clarity—to help shape AI systems that genuinely complement classroom needs. Our work establishes the technical foundation for this evolution, while future extensions must bridge the gap between computational correctness and instructional effectiveness.

## 6    LIMITATIONS

**Educational Focus and Scope.**    PCEVAL prioritizes educational contexts, focusing on introductory-level projects common in STEM classrooms. While valuable for teaching applications, this scope excludes professional physical computing scenarios involving advanced constraints (e.g., power optimization, real-time requirements, EMI considerations). Therefore, our findings primarily inform educational AI tools rather than industrial or research-grade embedded systems.

**Platform Scope and Generalizability.**    We focus on Arduino Uno due to its prevalence in introductory physical computing curricula (El-Abd, 2017; García-Tudela & Marín-Marín, 2023; Schätz et al., 2024). While this narrows the scope, PCEVAL's task decomposition (logical/physical circuit and code generation) and component-agnostic I/O specifications are platform-independent by design. To empirically check portability, we conducted a cross-platform validation using ESP32 microcontroller; automated checks confirmed equivalent functional behavior without changes to tasks or metrics (Appendix G).

## 7    CONCLUSIONS

We introduced PCEVAL, a benchmark for systematically evaluating LLM capabilities in physical computing tasks. Our evaluation of 13 models reveals that while LLMs perform reasonably well in code generation and logical circuit design, they struggle significantly with physical circuit generation, particularly in managing pin conflicts and breadboard layout constraints. This performance gap indicates that current LLMs have difficulty reasoning about spatial and physical implementation details that are essential for educational physical computing. While mitigation strategies like self-improvement and chain-of-thought prompting provided partial improvements, the challenges remain most pronounced in tasks requiring precise pin assignments and adherence to breadboard mechanics—areas where students typically need the most guidance. Educational validation through focus groups confirmed that effective classroom deployment requires consideration of pedagogical factors beyond technical correctness, including step-by-step guidance and visual clarity. These findings suggest that AI support tools for physical computing education must balance automation with learning objectives.

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

## A  INTERVIEWS WITH COMPUTER SCIENCE EDUCATORS

To gain comprehensive insights into the practical challenges encountered in physical computing education, interviews were conducted with eight educators. This group included six in-service computer science teachers from public middle and high schools (Teacher A, B, C, D, E, F) and two instructors with experience in private educational settings (Instructor G, H). The interviews aimed to identify common difficulties in areas such as class preparation, managing diverse student needs, providing individualized support, the inherent benefits and drawbacks of physical computing, and the need for better tools and resources. All educator names have been anonymized. Each interview lasted approximately 30 minutes.

### A.1  KEY INTERVIEW QUESTION AREAS

- **Q1:** What are the primary challenges and time-consuming aspects encountered when preparing for and conducting physical computing or practical coding classes?
- **Q2:** How do diverse student skill levels and the need for individualized support impact the flow of physical computing classes, and what difficulties arise in providing this support?
- **Q3:** What are the perceived benefits and drawbacks of physical computing classes, particularly regarding hardware integration (like Arduino) with software?
- **Q4:** What kind of additional support, tools, or resources do educators feel are needed to conduct physical computing classes more effectively, including the potential role of LLMs?

### A.2  THEMATIC SUMMARY OF INTERVIEW RESPONSES WITH EXCERPTS

**Q1:** What are the primary challenges and time-consuming aspects encountered when preparing for and conducting physical computing or practical coding classes?

> Educators consistently highlighted the significant time investment required for both preparing and conducting physical computing classes. Key challenges include the initial setup of hardware and software, creation or sourcing of suitable teaching materials, and managing the wide range of student skill levels.

- **Teacher A**: "Well, I think all teachers would agree that the most difficult stage is the connection stage." "Each time, firmware and installed programs disappear, which is a challenge." "So, I think it's not easy for teachers to change teaching aids every year." "During class, well, connection is something I prepare beforehand, but we always connect when class starts. So, in a way, that causes delays in the class progression."
- **Teacher B**: "First, the initial setup takes longer than expected." "But just as students need time to get used to new equipment or software they are handling for the first time, understanding the machine or physical computing device and how to connect it takes quite some time, which often takes longer than the essential programming part." "But for physical computing, the order and structure need to be slightly rearranged by the teacher depending on the task."
- **Teacher C**: "First, when the practical exercise begins, I try to create reference materials that are as detailed as possible for the students." "And when we actually start, many unexpected errors appear. Among them are the students' lack of experience with circuits, and also the stability of the boards." "Compatibility boards often perform poorly. So, I always prepare plenty of spares." "Driver installation and such, for compatibility boards, sometimes require installing separate drivers. There are many such environmental factors. And if students bring their own laptops, the number of things to set up increases significantly." "So, initially, when we did the camp with the Digital Sprout (CS Camp) [Anonymous] University side, even following the manuals they sent, files wouldn't upload at all... I remember digging through English manuals, manually setting everything up, and working on that continuously for a week before the class."
- **Teacher E**: "Then, the initial setup actually takes the longest time. In the case of Micro:bit, the setup environment doesn't take that extremely long, but anyway, when using any new teaching aid, I anticipate it will probably take a very long time." "Even simple things like having to sign up for a new environment, each little thing becomes a hurdle."
- **Teacher F**: "In the case of specialized high schools, since I'm not teaching the subjects I expected, I have to prepare for classes anew every time." "The content isn't light either, so I have to study while preparing for class, and I also have to create materials." "However, for specialized high school subjects, there aren't many materials, so I think the biggest difficulty is that I have to create everything myself." "Even in those, as I mentioned earlier, since class materials are not widely shared, I think creating examples for the students to do is the most difficult part."
- **Instructor G**: "Yes, robot coding was quite difficult. Since I can't use the robot at home, it's an item that's only at the academy. So even if I studied a lot before going, there were times when it didn't work well in practice." "So, I really wanted to go to the academy a bit late, but once I went about an hour early to try it out myself."
- **Instructor H**: "The step that took a lot of time in the preparation process was that we usually had to continuously check the mini-cars for any malfunctions or charging status before and after class." "Also, for auxiliary teaching aids like partitions or obstacles, we had to continuously check if there were any issues, and that process seemed to take a lot of time."

**Q2:** How do diverse student skill levels and the need for individualized support impact the flow of physical computing classes, and what difficulties arise in providing this support?

A major recurring theme was the challenge of managing diverse student abilities. Instructors find it difficult to cater to both fast learners and those who struggle, often leading to class delays and an

inability to provide equal support to all students. Troubleshooting individual student issues, whether hardware or software, consumes significant class time.

- **Teacher A**: "Individual understanding varies much more with these teaching aids. So, I have to walk around a lot. (To support students)" "If a connection is lost mid-class, that team gets delayed, and all other teams have to wait for that team to reconnect." "I think students ask more questions when their understanding of the tool is lacking." "Yes, yes, yes, I definitely think so [that there isn't enough time for one teacher to help all students]." "I think the maximum number of students one teacher can handle is about 15." "If all four members lack aptitude or are all struggling and fall behind, I end up focusing on those students and can't provide much feedback to other teams. The other teams might have finished and are just playing around."

- **Teacher B**: "Even with the same code, the sensors and other physical computing components differ slightly, so things often don't operate as desired." "Motors might suddenly break down, sensors might fail, or physical connections might come loose. When such things happen, spending time addressing them means less time for the actual lesson... losing a lot of time that way happens frequently." "The variation among students is also very large, so it's always good to prepare one or two advanced tasks or tasks that students can do according to their level."

- **Teacher C**: "Once you exceed about 10 students, practical physical computing classes become very difficult." "First, students find building circuits quite challenging... many kids lack a high understanding of electrical circuits... Then, we also had to check if the code was written correctly accordingly, so that was a somewhat difficult part." "When the friend next to them has it working, but theirs isn't... their sense of achievement can diminish quickly... feedback needs to be fast, there need to be many people available to help, and peer learning... need to be utilized much more." "Over 60% [of class time is spent giving feedback]."

- **Teacher D**: "Since the class pace is extremely fast, students who grasp things well and those who already have some basic knowledge... follow along well. But students who are encountering the language for the first time seem to take some time to apply concepts and write code. So, there's quite a bit of difference in level." "It's rather a very small minority, so those kids can't confidently ask for help like this. During class time." "If you start helping one student with their question, you have to keep helping that student, and then you can't cover all the material."

- **Teacher E**: "First, the students have a very hard time keeping up. Initially, when using something new, they really struggle." "I do one step, then go around and check everyone, then do another step, go around and check everyone again. This keeps repeating... eventually, I have to go around and check on everyone." "The class gets significantly delayed."

- **Teacher F**: "The biggest thing is the difference in level among the students." So, some students finish too quickly and are playing around after about 10 minutes, while some students don't understand until the very end, so I have to stand next to them and help them continuously." "As I explain it more simply and easily, the time spent with one student becomes longer. So, the overall class time becomes loose.'

- **Instructor G**: "The students' levels are really so different." "To match their levels, I tried many activities. First, I tried teaching them one-on-one... but that was too hard for me." "Because the students' levels were so different, if the number of students increased, it seemed very difficult to match each student's progress or solve each student's problems."

- **Instructor H**: "Some students who didn't listen properly to the theory class didn't understand the meaning or main activity of the practical exercise. So, for these students, we had the difficulty of having to explain the theory class content again while helping them with the practical activity." "Since there were 15 teams of middle school students, there were parts that students didn't understand,

and also students who couldn't proceed properly, so we had to help them one by one. There were several times when I felt that one main instructor and two assistant instructors were not enough to support 15 teams." "Explaining the theory to each team, explaining what the practical content was, and helping them... usually meant that the practical exercise time... would often exceed 1 minute per team. Therefore, there was a shortage of time to guide all 15 teams."

**Q3:** What are the perceived benefits and drawbacks of physical computing classes, particularly regarding hardware integration (like Arduino) with software?

Educators see significant value in physical computing for its ability to engage students and make abstract concepts tangible. However, they also point out the inherent difficulties in debugging hardware, managing unpredictable component behavior, and the financial and maintenance burden of hardware kits.

- **Teacher A**: "Advantages, well, the kids definitely enjoy it. I think it's one of the classes they find most fun." "they can actually see it with their own eyes, and I think that generates a lot of interest." "The disadvantage... Students who find it difficult find this more challenging than programming." "First, many students struggle with the connection stage." "If they've written the code perfectly, but it doesn't run as expected, it could actually be a flaw in the physical computing tool itself... I think the difficulty also lies in having to consider two things [hardware and software]."
- **Teacher B**: "For classes using physical computing, kids often only work in virtual environments like Python, so they don't know how it's actually used... But when they actually see something move... I think they feel, 'Ah, this is why it's necessary.'" The biggest disadvantage is, firstly, it's not easy for schools to provide the budget for this." "Secondly... managing it continuously is difficult. Especially for Arduino, sensors and similar items are often disposable."
- **Teacher C**: "The advantage is that it seems to foster interest in the various electrical appliances and many tools in our actual daily lives." "The problem is, they need continuous success to maintain interest." "As for disadvantages, it consumes too much time [for teacher preparation]." "Acquiring or purchasing such hardware in Korea is quite cumbersome... things sold as kits, are often too expensive."
- **Teacher E**: "The biggest advantage seems to be capturing the students' concentration and interest." "However, the disadvantages could be that teaching aids can break, and that students get too distracted by them and don't focus properly on the lesson." "First, there's no equipment. Primarily, there's no equipment." "Last year, our school had absolutely no budget for the computer science subject."
- **Teacher F**: "The biggest advantage is that it's not just theory; students can try things out and internalize them." "The disadvantage seems to be... it takes a lot of time, and the direction I planned for the class and the direction the students actually follow are different." "And the biggest disadvantage in terms of teaching aids is that they cost a lot of money."
- **Instructor G**: "In the case of robot coding, using LEGOs definitely sparks students' interest. So, because it's fascinating, they become more immersed, more focused, and more participative." "[Disadvantage:] If students drop them, it seems to result in a very large financial loss." "If the number of LEGOs for robot coding we are using is limited, if it breaks or an error occurs, there have been cases where that student could hardly proceed with the class."
- **Instructor H**: "Classes using teaching aids... have the advantage of consolidating the theoretical content once more and allowing students to understand it better by directly experiencing the theory with their hands." "When students use physical teaching aids... their concentration level increases, and they don't get bored." "The biggest disadvantage... is that since it's an activity conducted after

the theory class, for students who don't understand the theoretical content, both the theory class and the teaching aid class feel rather meaningless." "There were students who would damage the teaching aids, such as by pressing that sensor part with their hands, scribbling on the teaching aids, or trying to break them."

**Q4:** What kind of additional support, tools, or resources do educators feel are needed to conduct physical computing classes more effectively, including the potential role of LLMs?

There's a clear call for better support systems, including more accessible and detailed guidelines for students, improved teacher training, larger budgets for hardware, and tools that can help bridge the gap between software and hardware. The potential of LLMs is recognized, but their current limitations in addressing physical hardware challenges are also noted.

- **Teacher A**: "I wish there were more opportunities for the creators of the tools, or for places that professionally train teachers on these tools, to provide regular training." "Sharing of teaching materials that teachers can use easily." "The connection methods need to be developed to be easy, and those developed connection methods should be widely distributed so that students can understand them at their level." "[Regarding LLMs for students] First-year middle school students can't use GPT, so I've never tried it, and their Googling skills are also lacking... So, if there were a platform or something that summarized this kind of information, I would refer to that platform."

- **Teacher B**: "Students learn Python, but to actually learn Arduino, they might need to be taught C++, which presents a double cognitive load. So, finding a way to reduce that seems to be the first necessity." "Secondly, many teachers perceive physical computing as difficult... I think teacher training or similar initiatives are needed to reduce those perceptions." "For Arduino, yes, there have been times like that [using ChatGPT]. but... often they just end up asking 'make it for me.'... a lack of understanding of the coding itself tends to occur."

- **Teacher C**: "Feedback needs to be fast, there need to be many people available to help, and peer learning... need to be utilized much more." "For the code parts, naturally, these days... using various AI like ChatGPT, you can catch errors, get help with code writing for connections, etc. But in the physical aspect, giving students quick feedback on various errors is really necessary. Especially in physical computing, the circuit part is crucial, and this is something AI absolutely cannot do for you... An arm doesn't suddenly pop out from there to build the circuit." "Nowadays, like Collab, Jemaine just wrote the code and recommended it to me, and it's like this." "I wish schools had more budget to purchase such things." "To do physical computing, you need a small makerspace-like area... but those are often lacking."

- **Teacher D**: "[When using LLMs for class preparation] I still don't think the quality and completeness are high enough to be immediately used in practice... Since it can't be used directly, I figure it's easier to just quickly make the blanks myself." "[When do I use GPT?] When showing short code examples for practice during class."

- **Teacher E**: "First, there's no equipment. Primarily, there's no equipment." "Last year, our school had absolutely no budget for the computer science subject." "The entire budget for teaching aid purchase was 1 million won."

- **Teacher F**: "First, I hope the Ministry of Education provides a lot of budget." "I think it would be good if there was a tool or service that could organize information about such teaching aids or share it with teachers." "[Regarding LLMs] since I can't code, I always use GPT to search."

- **Instructor G**: "I think the guidelines given to students are too few." "If we could distribute guidelines to each student on how they can actually use them... if detailed guidelines were available through a platform on those laptops, it would be a huge help during class." "If they could model what they're building with their hands at home before coming to the academy, if they could try to fit the pieces together and implement it in software beforehand, it would be a huge help."
- **Instructor H**: "In the case of teaching aids, malfunctions are frequent, and I believe that continuously developing new teaching aids is more helpful for students' classes. Therefore, I think financial support is obviously necessary." "If there was a management system or a communication tool, instructors could communicate better, and it would be convenient to have a community where we can report and inform each other about problematic teaching aids or tools."

Table 5: Comprehensive circuit validation results showing average error frequency across all evaluated models, categorized by error type and circuit representation.

| Model | Circuit Type | Logical / Physical Errors | | | | Physical Only Errors | |
|---|---|---|---|---|---|---|---|
| | | Redundant Connection | Extraneous Component | Missing Component | Isolated Component | Pin Conflict | Breadboard Bypass |
| GPT-4o-mini | Logical | 0.01 | 1.19 | 0.26 | 0.29 | - | - |
| | Physical | 0.01 | 0.18 | 0.23 | 0.61 | 4.49 | 5.15 |
| Claude 3.5 Haiku | Logical | 0.08 | 0.06 | 0.30 | 0.07 | - | - |
| | Physical | 0.07 | 0.68 | 0.13 | 0.50 | 3.56 | 5.60 |
| Gemini-2.0-Flash-Lite | Logical | 0.16 | 0.56 | 0.03 | 0.35 | - | - |
| | Physical | 0.20 | 0.63 | 0.03 | 0.21 | 5.18 | 3.72 |
| GPT-4.1 | Logical | 0.0 | 0.03 | 0.03 | 0.03 | - | - |
| | Physical | 0.27 | 0.18 | 0.0 | 0.12 | 7.27 | 0.02 |
| Gemini-2.0-Flash | Logical | 0.0 | 0.12 | 0.09 | 0.09 | - | - |
| | Physical | 0.0 | 0.46 | 0.13 | 0.32 | 3.53 | 2.73 |
| Claude 3.7 Sonnet | Logical | 0.0 | 0.33 | 0.02 | 0.02 | - | - |
| | Physical | 0.0 | 1.45 | 0.0 | 1.15 | 7.52 | 0.17 |
| GPT-4o | Logical | 0.0 | 0.08 | 0.08 | 0.03 | - | - |
| | Physical | 0.03 | 0.20 | 0.01 | 0.51 | 2.07 | 1.16 |
| o3-mini | Logical | 0.02 | 0.39 | 0.06 | 0.02 | - | - |
| | Physical | 0.0 | 0.21 | 0.02 | 0.06 | 4.20 | 0.01 |
| Gemma 3 | Logical | 0.09 | 0.67 | 0.30 | 0.58 | - | - |
| | Physical | 0.0 | 0.32 | 0.15 | 0.90 | 3.63 | 1.26 |
| Phi 4 | Logical | 0.0 | 0.03 | 0.03 | 0.03 | - | - |
| | Physical | 0.01 | 0.13 | 0.39 | 0.21 | 3.47 | 2.94 |
| Mistral-small | Logical | 0.45 | 0.75 | 0.13 | 0.03 | - | - |
| | Physical | 0.59 | 0.01 | 0.19 | 0.04 | 2.35 | 1.01 |
| Deepseek-Coder-v2 | Logical | 0.05 | 0.33 | 0.60 | 0.53 | - | - |
| | Physical | 0.02 | 0.09 | 1.69 | 0.08 | 0.53 | 3.93 |
| LLaMA3.1 | Logical | 0.07 | 0.42 | 0.73 | 0.45 | - | - |
| | Physical | 0.04 | 0.1 | 0.44 | 0.07 | 1.15 | 2.23 |

Table 6: CodeBLEU quality assessment of LLM-generated code, comparing scores between successful and unsuccessful simulation outcomes for different code generation tasks.

| Model | $D, L \rightarrow C$ | | $D, P \rightarrow C$ | |
| --- | --- | --- | --- | --- |
| | **CodeBLEU** | | **CodeBLEU** | |
| | **Success** | **Failure** | **Success** | **Failure** |
| GPT-4o-mini | 0.48 | 0.38 | 0.47 | 0.37 |
| Claude 3.5 Haiku | 0.49 | 0.41 | 0.48 | 0.43 |
| Gemini-2.0-Flash-Lite | 0.55 | 0.47 | 0.53 | 0.47 |
| GPT-4.1 | 0.53 | 0.52 | 0.52 | 0.50 |
| Gemini-2.0-Flash | 0.54 | 0.49 | 0.57 | 0.47 |
| Claude 3.7 Sonnet | 0.52 | 0.46 | 0.51 | 0.46 |
| GPT-4o | 0.52 | 0.45 | 0.53 | 0.43 |
| o3-mini | 0.50 | 0.43 | 0.48 | 0.44 |
| Gemma 3 | 0.50 | 0.42 | 0.56 | 0.34 |
| Phi 4 | 0.47 | 0.42 | 0.49 | 0.36 |
| Mistral-Small 3 | 0.59 | 0.48 | 0.59 | 0.38 |

## B  EDUCATIONAL VALIDATION THROUGH EXPERT FOCUS GROUPS

### B.1  PARTICIPANTS AND PROCEDURE

We conducted two semi-structured interviews with physical computing educators (8 and 10 years of teaching experience, respectively; 30 minutes each) and collected responses from three pre-service CS teachers (senior undergraduates who had completed teacher certification) via a short online survey.

### B.2  KEY FINDINGS FROM EDUCATORS

- **Step-by-step guidance:** Educators emphasized that students require sequential assembly instructions (e.g., "now connect the red wire to pin 13"), which current LLM outputs do not provide.

- **Multiple candidate solutions:** Presenting alternative layouts was highlighted as pedagogically valuable for teaching breadboard principles, compared to providing only one "answer."

- **Readability principles:** Practical design habits (e.g., consistent wire colors, left-oriented layouts) were noted as essential for teaching, yet absent from current benchmarks.

- **Reducing capability gaps:** Educators noted that LLMs could help less-experienced teachers identify circuit problems and provide richer feedback to students.

- **Validation of automated simulation:** Simulation-based verification was recognized as a major strength, reducing teacher workload by ensuring functional correctness before classroom use.

### B.3  FINDINGS FROM PRE-SERVICE TEACHERS

Survey participants rated educational value across project complexity levels:

- Low-complexity projects: Readability 4.67/5, Correctness 5.0/5, Educational Value 4.0/5.

- Medium-complexity: Readability 1.67/5, Correctness 4.67/5, Educational Value 2.0/5.

- High-complexity: Readability 1.0/5, Correctness 2.33/5, Educational Value 1.0/5.

They consistently identified excessive jumper wires, convoluted routing, and overlapping components as barriers to classroom usability.

### B.4 IMPLICATIONS

These findings confirm that while functional correctness is a prerequisite, true educational utility also requires readability, process guidance, and layout simplicity. PCEVAL provided the systematic framework to reveal these limitations, highlighting directions for future benchmarks and LLM evaluation methods.

## C EVALUATION METRIC DETAILS

This appendix details the metrics used to evaluate the outputs generated by LLMs in the PCEVAL benchmark. We assess both the structural integrity of generated circuits and the quality of the generated code, providing a comprehensive view of LLM capabilities in physical computing tasks.

### C.1 SIMULATOR

Reference implementations were validated across real hardware and multiple simulators (Wokwi, Tinkercad, Virtual Breadboard, SimulIDE), ensuring cross-platform consistency. We ultimately selected Wokwi for automated evaluation due to its API accessibility and component coverage, while the framework itself remains simulator-agnostic.

### C.2 CIRCUIT VALIDATION

Our circuit validation protocol identifies structural errors in both logical and physical circuit designs generated by LLMs. This involves six distinct error checks:

Four metrics apply to both logical and physical circuits:

- **Redundant Connection:** Duplicate connections between the same two component pins (e.g., listing both [A, B] and [B, A]).

- **Extraneous Component:** A component included in the LLM-generated circuit that is not present in the ground truth (reference) circuit.

- **Missing Component:** A component present in the ground truth circuit but omitted from the LLM-generated circuit.

- **Isolated Component:** A component correctly included in the generated circuit but lacking any necessary electrical connections.

For physical circuit evaluation, two additional metrics specifically address implementation constraints on a breadboard:

- **Pin Conflict:** Assigning multiple connections to a single, indivisible breadboard hole or a component pin that cannot accept multiple direct connections.

- **Breadboard Bypass:** Creating direct pin-to-pin connections between components without utilizing the breadboard, violating standard physical prototyping practices for the given tasks.

Table 5 presents the comprehensive circuit validation results, showing the average frequency of these errors across all evaluated models and circuit types.

Table 7: Full results of self-improvement experiments (success rates, %).

| Model | Score | $D, L \rightarrow C$ | $D, P \rightarrow C$ | $D, C \rightarrow L$ | $D, C \rightarrow P$ |
|---|---|---|---|---|---|
| o3-mini | $65.3 \rightarrow 76.5$ | $71.6 \rightarrow 83.6$ | $71.6 \rightarrow 84.8$ | $70.0 \rightarrow 79.6$ | $48.0 \rightarrow 58.0$ |
| Gemini-2.0 Flash | $49.0 \rightarrow 57.2$ | $58.0 \rightarrow 72.8$ | $54.4 \rightarrow 66.0$ | $62.4 \rightarrow 73.2$ | $21.2 \rightarrow 16.8$ |

Table 8: Full results of CoT prompting experiments (success rates, %).

| Model | $D, P \rightarrow C$ | $D, C \rightarrow P$ |
|---|---|---|
| Mistral-Small 24B | $30.8 \rightarrow 48.8$ | $13.6 \rightarrow 11.6$ |
| GPT-4o | $56.4 \rightarrow 66.8$ | $26.8 \rightarrow 29.6$ |
| GPT-4o-mini | $51.2 \rightarrow 52.0$ | $1.2 \rightarrow 2.4$ |
| Phi-4 | $35.6 \rightarrow 47.2$ | $2.8 \rightarrow 3.2$ |
| LLaMA 3.1 | $24.0 \rightarrow 22.4$ | $2.0 \rightarrow 5.2$ |
| Gemma 3 | $28.4 \rightarrow 45.2$ | $2.4 \rightarrow 2.8$ |
| DeepSeek Coder V2 | $20.8 \rightarrow 32.4$ | $1.2 \rightarrow 0.0$ |

## C.3 CODE QUALITY VALIDATION

Beyond functional correctness determined by simulation success, we assess the quality of LLM-generated Arduino code using CodeBLEU (Ren et al., 2020). Table 6 presents the CodeBLEU scores for both code generation tasks ($D, L \rightarrow C$ and $D, P \rightarrow C$), comparing outputs that led to simulation success versus those that failed.

An interesting observation from these scores is that functionally successful code did not always achieve substantially higher CodeBLEU scores than code that failed simulation. In some instances, the scores were comparable (e.g., GPT-4.1 on $D, L \rightarrow C$: Success 0.53, Failure 0.52; o3-mini on $D, P \rightarrow C$: Success 0.48, Failure 0.44). This phenomenon may suggest that LLMs can generate code that is structurally or syntactically similar to the correct solution, thus achieving a comparable CodeBLEU score, yet still fail functionally due to subtle but critical errors, such as incorrect pin number assignments. Such errors might not significantly penalize the CodeBLEU score, which primarily assesses aspects like n-gram matching, syntactic correctness, and dataflow similarity, but are critical for the successful execution of physical computing projects.

## D ADDITIONAL RESULTS: SELF-IMPROVEMENT AND CHAIN-OF-THOUGHT

Tables 7 and 8 present the complete results for our self-improvement and CoT prompting experiments. As discussed in Section 4.3, the main trends are consistent across models: (1) self-improvement yields robust and steady performance gains, especially in code-related tasks, while (2) CoT exhibits more variable benefits, occasionally even degrading performance on hardware tasks.

## E PCEVAL BENCHMARK PROJECTS

This appendix provides a detailed overview of all projects included in the PCEval benchmark dataset. The projects are categorized by complexity level, outlining the specific tasks and hardware components involved in each.

While the absolute number of projects (50) may appear modest in isolation, the design of PCEVAL ensures a sufficiently large and diverse evaluation space. Each project supports four distinct generation tasks, resulting in 200 unique evaluation samples. This scale is comparable to established software benchmarks such as HumanEval, which includes 164 problems for code-only tasks (Chen et al., 2021). In physical computing, dataset construction is substantially more complex than in code-only settings, since each instance requires not only executable code but also consistent logical and physical circuit specifications, validated across both simulation environments and real hardware.

The most directly related prior work, MICRO25 (Jansen, 2023), introduced 25 projects with natural language descriptions. However, MICRO25 provides no reference implementations (code or circuits) and relies entirely on manual expert evaluation, limiting reproducibility and scalability. In contrast, PCEVAL doubles the number of projects (50), and each project includes complete reference implementations—code, logical circuits, and physical circuits—supporting fully automated and reproducible evaluation across 200 task instances.

Table 9: Overview of PCEval Benchmark Projects by Level

| Level | Project Name | Description |
|---|---|---|
| Level 1 | 7 segment display basic | Connect a 7-segment display to Arduino and write code to show the number 5. |
| | bar led basic | Connect a bar graph LED to Arduino and write code to turn on LEDs 1 through 6. |
| | buzzer basic | Connect a piezo buzzer to Arduino and write code to play a sound. |
| | distance sensor basic | Connect an ultrasonic distance sensor to Arduino and write code to measure the distance and print it to the serial monitor. |
| | humidity sensor basic | Connect a DHT22 sensor to Arduino and write code to measure the humidity and print it to the serial monitor. |
| | LCD display basic | Connect an ILI9341 LCD display to Arduino and write code to show "Hello World!" on the display. |
| | led blink basic | Connect an LED to Arduino and write code to repeatedly turn the LED on and off with delays. |
| | led RGB basic | Connect an RGB LED to Arduino and write code to turn on only the blue segment. |
| | photoresistor basic | Connect a photoresistor sensor module to Arduino and write code to detect light intensity and print it to the serial monitor. |
| | RTC module basic | Connect an RTC module to Arduino and write code to read the current time and print it to the serial monitor. |
| | servo motor basic | Connect a servo motor to Arduino and write code to repeatedly move the servo to 0° and 180°. |
| | temperature sensor basic | Connect a DHT22 sensor to Arduino and write code to measure the temperature and print it to the serial monitor. |
| Level 2 | 7 segment display counter | Connect a 7-segment display to Arduino and write code to show 1, 2, and 3 based on elapsed time. |
| | accelerometer | Connect an MPU6050 accelerometer to Arduino and write code to measure acceleration on the X, Y, and Z axes and print the values to the serial monitor. |

**Table 9 – continued from previous page**

| Level | Project Name | Description |
|---|---|---|
| | button duration | Connect a push button to Arduino and write code to measure how long the button is pressed and print the duration to the serial monitor. |
| | button pulldown | Connect a push button to Arduino with a pulldown resistor and write code to monitor the button state and print it to the serial monitor. |
| | button pullup | Connect a push button to Arduino with a pullup resistor and write code to monitor the button state and print it to the serial monitor. |
| | distance sensor | Connect an ultrasonic distance sensor to Arduino and write code to measure the distance and compare it with a threshold. |
| | gyroscope | Connect an MPU6050 gyroscope to Arduino and write code to measure rotation on the X, Y, and Z axes and print the values to the serial monitor. |
| | humidity sensor | Connect a DHT22 sensor to Arduino and write code to measure the humidity and compare it with a threshold. |
| | serial bar led | Connect a bar graph LED to Arduino and write code to read serial input and light up the corresponding number of LEDs. |
| | serial LCD display | Connect an ILI9341 LCD display to Arduino and write code to read serial input and display it on the LCD. |
| | serial monitor | Connect nothing to Arduino and write code to read serial input and print it back to the serial monitor. |
| | serial RGB led | Connect an RGB LED to Arduino and write code to read serial input and turn on the corresponding RGB segment. |
| | temperature sensor | Connect a DHT22 sensor to Arduino and write code to measure the temperature and compare it with a threshold. |
| Level 3 | 4 digit 7 segment display | Connect a 4-digit 7-segment display to Arduino and write code to show 1, 2, and 3 based on elapsed time. |
| | 7 segment display serial | Connect a 7-segment display to Arduino and write code to show serial input on the display. |
| | button buzzer | Connect a push button and piezo buzzer to Arduino and write code to play a sound when the button is pressed. |
| | button LCD display | Connect a push button and ILI9341 display to Arduino and write code to show the button state on the LCD. |
| | button led | Connect a push button and LED to Arduino and write code to turn on the LED when the button is pressed. |
| | button RGB led | Connect three push buttons (red, green, blue) and an RGB LED to Arduino and write code to turn on the corresponding color segment when each button is pressed. |
| | button RTC timezone | Connect a push button and RTC module to Arduino and write code to print the time in different timezones based on the button state. |
| | button servo motor | Connect a push button and servo motor to Arduino and write code to change the servo motor angle using the button. |

**Table 9 – continued from previous page**

| Level | Project Name | Description |
|---|---|---|
| | dht22 LCD display | Connect a DHT22 sensor and an ILI9341 LCD display to Arduino and write code to show humidity and temperature values on the LCD. |
| | multiplexer photoresistor | Connect six photoresistors to Arduino via a 16-channel analog multiplexer and write code to print each sensor's value. |
| | multiplexer potentiometer | Connect five potentiometers to Arduino via a 16-channel analog multiplexer and write code to print each potentiometer's value. |
| | photoresistor bar led | Connect a bar graph LED and photoresistor to Arduino and write code to light up the bar LEDs based on the mapped value of the photoresistor. |
| | potentiometer bar led | Connect a bar graph LED and potentiometer to Arduino and write code to light up the bar LEDs based on the mapped value of the potentiometer. |
| | potentiometer servo motor | Connect a potentiometer and servo motor to Arduino and write code to set the servo motor angle based on the mapped potentiometer value. |
| Level 4 | alarm | Connect a piezo buzzer and RTC module to Arduino and write code to play a sound when the specified time is reached. |
| | binary led | Connect five LEDs to Arduino and write code to show the binary representation of serial input using the LEDs. |
| | calendar display | Connect an RTC module, an ILI9341 display, and two push buttons to Arduino and write code to show the current date on the LCD. Use the buttons to navigate between dates. |
| | clock | Connect a 4-digit 7-segment display and RTC module to Arduino and write code to show the current time in HH:MM format. |
| | dday counter | Connect an RTC module to Arduino and write code to read a date from serial input and print the number of days remaining until that date. |
| | exercise counter | Connect an MPU6050 sensor and 7-segment display to Arduino and write code to count each time the Y-axis rotation exceeds a threshold, then show the count on the display. |
| | multiple timezone | Connect an RTC module and four push buttons to Arduino and write code to display different timezones depending on which button is pressed. |
| | parking spot monitor | Connect an ultrasonic distance sensor and RGB LED to Arduino and write code to measure distance and switch from green to red light if the distance is below a threshold. |
| | piano keyboard | Connect four push buttons and four piezo buzzers to Arduino and write code to play a tone when the corresponding button is pressed. |
| | smart led system | Connect five LEDs and five photoresistors to Arduino and write code to turn on each LED when its corresponding photoresistor reads below a threshold. |

**Table 9 – continued from previous page**

| Level | Project Name | Description |
|-------|-------------|-------------|
| | traffic light | Connect a push button and three LEDs to Arduino and write code to turn on the green LED by default, then switch to yellow and red in order when the button is pressed. |

## F  PROMPT DETAILS

This appendix provides the specific prompts used for each of the four core generation tasks in the PCEVAL benchmark. Each prompt was carefully designed to clearly articulate the task objectives to the LLMs and to specify the required output format. For tasks involving physical circuit generation or code generation from physical circuits, prompts included detailed explanations of breadboard mechanics and electrical connectivity rules to ensure LLMs had the necessary contextual information.

Code Generation Task (Logical Hardware)

```
# Arduino Code Generation Task (Logical Hardware)

## Task
Please generate Arduino code (main.ino) that will work with this logical
hardware configuration and produce the expected behavior.
The code should fulfill all requirements and pass all test steps in the
scenario.

Note that this is a LOGICAL circuit diagram, meaning it shows the direct
connections between components at a conceptual level, NOT physical layout
with breadboard positions.

### Output Format
Please provide only the code without explanations or markdown formatting.

## Project Description
{Project Description Text}

## Hardware Configuration (Logical Circuit)
Below is the JSON diagram describing the logical hardware circuit:
```json
{Standardized Logical Diagram JSON}
```
```

Code Generation Task (Physical Hardware)

```
# Arduino Code Generation Task (Physical Hardware)

## Task
```

```
Please generate Arduino code (main.ino) that will work with this physical
hardware configuration and produce the expected behavior.
The code should fulfill all requirements and pass all test steps in the
scenario.

Note that this is a PHYSICAL circuit diagram, meaning it shows the actual
breadboard layout with specific component placements and wire connections
using breadboard positions.

### Output Format
Please provide only the code without explanations or markdown formatting.

## Project Description
{Project Description Text}

## Hardware Configuration (Physical Breadboard Layout)

### Pin and Position Naming Conventions
- Component pins: "arduino1.pin13", "led1.anode", "resistor1.pin1", etc.
- Breadboard positions (main area): "breadboard.COLUMN_ROW"
    - Column: A number from 1-60
    - Row: A letter from a-j
    - Rows a-e are on the top half, rows f-j are on the bottom half
    - Example: "breadboard.10a" refers to column 10, row a (top half)
- Breadboard power rail positions: "breadboard.RAIL_TYPE.COLUMN_NUMBER"
    - RAIL_TYPE: tp (top positive), tn (top negative), bp (bottom positive)
    , bn (bottom negative)
    - COLUMN_NUMBER: A number, typically referring to segments or positions
     along the rail (e.g., 1 up to 50)
    - Example: "breadboard.tp.1" refers to position 1 on the top positive
    (+) rail
    - Example: "breadboard.bn.12" refers to position 12 on the bottom
    negative (-) rail

### Breadboard Explanation
A breadboard is divided into two halves separated by a central gap:
- Top half: 5 rows of holes, commonly labeled a, b, c, d, e
  (from top to bottom for each column segment).
- Bottom half: 5 rows of holes, commonly labeled f, g, h, i, j
  (from top to bottom for each column segment).
- The columns are typically numbered 1-60 from left to right.
  So, for each column number, there's a set of 'a-e' holes and
  a set of 'f-j' holes.

#### Electrical Connections:
- Terminal Strips (the main area with rows a-e and rows f-j):
    - Within each half (top or bottom), and for each column number, the 5
    holes in that column segment are electrically connected vertically.
        - Example: `breadboard.1a`, `breadboard.1b`, `breadboard.1c`, `
        breadboard.1d`, and `breadboard.1e` are all connected.
        - Example: `breadboard.1f`, `breadboard.1g`, `breadboard.1h`, `
        breadboard.1i`, and `breadboard.1j` are all connected.
    - Holes in different columns are NOT connected. (e.g., `breadboard.1a`
    is not connected to `breadboard.2a`)
```

      - The top half (holes a-e) and the bottom half (holes f-j) are
      electrically separated by the central gap. (e.g., `breadboard.1e` is
      not connected to `breadboard.1f`)
    - Power Rails (tp, tn, bp, bn):
      - All holes in a single power rail (e.g., the 'tp' rail) are
      electrically connected horizontally along the length of that entire
      rail.
          - Example: All positions along "breadboard.tp" (e.g., `breadboard.
          tp.1`, `breadboard.tp.2`, ..., `breadboard.tp.50`) are connected.
      - Typically, the positive (+) and negative (-) rails are electrically
      separate from each other.
      - The top set of power rails (e.g., 'tp' and 'tn') are electrically
      separate from the bottom set of power rails (e.g., 'bp' and 'bn'),
      unless explicitly connected by external wires.

    Below is the JSON diagram describing the physical hardware circuit
    with breadboard layout:
    ```json
    {Standardized Physical Diagram JSON}
    ```

Circuit Generation Task (Logical)

    # Arduino Hardware Design Task

    ## Task
    Based on the Arduino code and project description, create a logical circuit
     diagram that will work correctly with this code.

    Focus on the direct connections between components at a conceptual level,
    NOT physical layout with breadboard positions.

    ## Output Format
    Provide your answer in JSON format with two main sections:
    1. "components": A list of all hardware components needed
    2. "connections": A list of all connections between components

    ### Components Format
    Each component should have:
    - "id": A unique, meaningful identifier with a number appended (e.g., "
    arduino1", "led2", "resistor1"), that is consistently used in the
    connections section
    - "type": Component type ("Arduino Uno", "LED", "Resistor", "Button", etc.)
    - "properties": Optional object with properties specific to the component
    type

    ### Connections Format
    Each connection is simply an array with exactly two elements, each being a
    connection point formatted as "componentId.pinName".
    For example: ["arduino1.pin13", "led2.anode"]

```
### Pin Naming Conventions
- Arduino pins: "pin2", "pin13", "a0", "a1", "5v", "3.3v", "gnd1", "gnd2",
"gnd3", etc.
- LED pins: "anode", "cathode"
- Resistor pins: "pin1", "pin2"
- Button pins: "pin1.l", "pin1.r", "pin2.l", "pin2.r"

### Example
```json
{
  "components": [
    {"id": "arduino1", "type": "Arduino Uno"},
    {"id": "ntc_sensor1", "type": "NTC temperature sensor",
     "properties": {"temperature": "24"}}
  ],
  "connections": [
    ["arduino1.pin13", "resistor2.pin1"],
    ["ntc_sensor1.VCC", "arduino1.5v"],
    ["ntc_sensor1.GND", "arduino1.gnd2"],
    ["ntc_sensor1.OUT", "arduino1.A0"]
  ]
}
```

Based on the code, provide a complete JSON with all components and direct
logical connections to create a functional circuit.

## Project Description
{Project Description Text}

## Arduino Code
```cpp
{Sketch Code Text}
```
```

Circuit Generation Task (Physical)

```
# Arduino Hardware Design Task

## Task
Based on the Arduino code and project description, create a physical
breadboard layout that will work correctly with this code.

Include a breadboard and show how components would be physically placed and
 connected on it.

## Output Format
Provide your answer in JSON format with two main sections:
1. "components": A list of all hardware components needed
2. "connections": A list of all connections between components
```

### Components Format
Each component should have:
- "id": A unique, meaningful identifier with a number appended  (e.g., "arduino1", "led2", "resistor1") that is consistently used in the connections section
- "type": Component type ("Arduino Uno", "LED", "Resistor", "Button", "Breadboard", etc.)
- "properties": Optional object with properties specific to the component type

### Connections Format
Each connection is simply an array with exactly two elements, which can be component pins or breadboard positions.
For example: ["arduino1.pin13", "breadboard1.10a"]

### Pin and Position Naming Conventions
- Component pins: "arduino1.pin13", "led1.anode", "resistor1.pin1", etc.
- Arduino pins: "pin2", "pin13", "a0", "a1", "5v", "3.3v", "gnd1", "gnd2", "gnd3", etc.
- LED pins: "anode", "cathode"
- Resistor pins: "pin1", "pin2"
- Button pins: "pin1.l", "pin1.r", "pin2.l", "pin2.r"
- Breadboard positions (main area): "breadboard.COLUMN_ROW"
    - Column: A number from 1-60
    - Row: A letter from a-j
    - Rows a-e are on the top half, rows f-j are on the bottom half
    - Example: "breadboard.10a" refers to column 10, row a (top half)
- Breadboard power rail positions: "breadboard.RAIL_TYPE.COLUMN_NUMBER"
    - RAIL_TYPE: tp (top positive), tn (top negative), bp (bottom positive), bn (bottom negative)
    - COLUMN_NUMBER: A number, typically referring to segments or positions along the rail (e.g., 1 up to 50)
    - Example: "breadboard.tp.1" refers to position 1 on the top positive (+) rail
    - Example: "breadboard.bn.12" refers to position 12 on the bottom negative (-) rail

### Breadboard Explanation
A breadboard is divided into two halves separated by a central gap:
- Top half: 5 rows of holes, commonly labeled a, b, c, d, e
  (from top to bottom for each column segment).
- Bottom half: 5 rows of holes, commonly labeled f, g, h, i, j
  (from top to bottom for each column segment).
- The columns are typically numbered 1-60 from left to right.
  So, for each column number, there's a set of 'a-e' holes and a set of 'f-j' holes.

#### Electrical Connections:
- Terminal Strips (the main area with rows a-e and rows f-j):
    - Within each half (top or bottom), and for each column number, the 5 holes in that column segment are electrically connected vertically.
        - Example: `breadboard.1a`, `breadboard.1b`, `breadboard.1c`, `breadboard.1d`, and `breadboard.1e` are all connected.
        - Example: `breadboard.1f`, `breadboard.1g`, `breadboard.1h`, `breadboard.1i`, and `breadboard.1j` are all connected.

```
        – Holes in different columns are NOT connected.
          (e.g., `breadboard.1a` is not connected to `breadboard.2a`)
        – The top half (holes a–e) and the bottom half (holes f–j) are
          electrically separated by the central gap.
          (e.g., `breadboard.1e` is not connected to `breadboard.1f`)
    – Power Rails (tp, tn, bp, bn):
        – All holes in a single power rail (e.g., the 'tp' rail) are
        electrically connected horizontally along the length of that entire
        rail.
            – Example: All positions along "breadboard.tp" (e.g., `breadboard.
            tp.1`, `breadboard.tp.2`, ..., `breadboard.tp.50`) are connected.
        – Typically, the positive (+) and negative (–) rails are electrically
        separate from each other.
        – The top set of power rails (e.g., 'tp' and 'tn') are electrically
        separate from the bottom set of power rails (e.g., 'bp' and 'bn'),
        unless explicitly connected by external wires.

### Example
```json
{
  "components": [
    {"id": "arduino1", "type": "Arduino Uno"},
    {"id": "breadboard1", "type": "Breadboard"},
    {"id": "ntc_sensor1", "type": "NTC temperature sensor",
     "properties": {"temperature": "24"}}
  ],
  "connections": [
    ["breadboard1.29g", "arduino1.gnd3"],
    ["ntc_sensor1.VCC", "breadboard1.28f"],
    ["ntc_sensor1.GND", "breadboard1.29f"],
    ["ntc_sensor1.OUT", "breadboard1.27f"],
    ["breadboard1.28g", "arduino1.5v"],
    ["breadboard1.27g", "arduino1.A0"]
  ]
}
```

### Important Constraints
– Each Arduino pin can only have ONE connection assigned to it.
– Each breadboard hole can only have ONE connection assigned to it.
– Direct pin–to–pin connections between components without using a
breadboard are not permitted for circuit construction involving a
breadboard. All such connections must be routed via breadboard positions.

Pay attention to the internal connections of the breadboard.
Based on the code, provide a complete JSON with all components and
connections to create a functional physical circuit with breadboard layout.

## Project Description
{Project Description Text}

## Arduino Code
```cpp
{Sketch Code Text}
```
```

Addressing these limitations in subsequent research will contribute to a more holistic understanding of LLM capabilities in the multifaceted domain of physical computing and further guide the development of AI tools that can support both novice learners and experienced practitioners.

# G ADDITIONAL EXPERIMENTS ON ESP32

To verify the platform portability of PCEVAL, we conducted additional experiments on ESP32, which is also a well-known physical computing platform similar to Arduino Uno. Among the projects conducted on Arduino, 24 projects were selected for experiments across four tasks, and they were evaluated under the same simulation environment (Shaked, 2020).

Table 10: Test results of LLM-generated code and circuit on the ESP32 platform, compared with Arduino scores. For the fairness, Arduino scores were computed solely from the subset of projects used in the ESP32 experiments.

| Model | $D, C \rightarrow L$ | | $D, C \rightarrow P$ | | $D, L \rightarrow C$ | | $D, P \rightarrow C$ | | Overall | |
|---|---|---|---|---|---|---|---|---|---|---|
| | Arduino | ESP32 | Arduino | ESP32 | Arduino | ESP32 | Arduino | ESP32 | Arduino | ESP32 |
| Gemini-2.0-Flash-Lite | 58.3 | **50.0** | 5.0 | **3.3** | 60.0 | **58.3** | 61.7 | **60.8** | 46.3 | **43.2** |
| Gemini-2.0-Flash | 71.7 | **57.5** | 30.0 | **11.7** | 72.5 | **62.5** | 70.0 | **56.7** | 61.1 | **47.1** |
| DeepSeek-Coder-v2 | 35.0 | **46.7** | 0.0 | **0.0** | 28.3 | **44.2** | 25.8 | **22.5** | 22.3 | **17.4** |
| LLaMA 3.1 | 27.5 | **27.5** | 1.7 | **0.8** | 30.8 | **38.3** | 28.3 | **29.2** | 22.1 | **24.0** |
| Phi 4 | 44.2 | **45.8** | 0.8 | **0.8** | 51.6 | **51.6** | 42.5 | **40.8** | 34.8 | **34.8** |
| Mistral-Small 3 | 57.5 | **55.0** | 13.3 | **14.2** | 56.7 | **50.8** | 40.8 | **45.0** | 42.1 | **41.2** |

Table 10 represents the test results of ESP32 code and circuit generated by six open and closed source LLMs. When comparing the ESP32 scores with those of Arduino, they showed clearly similar trends and values. This experimental results indicates that PCEval is not confined to the Arduino platform but can be extended to multiple platforms.

# H SYSTEM SPECIFICATIONS FOR EVALUATION

The primary evaluations of Large Language Models (LLMs) for the PCEVAL benchmark were conducted by querying their respective APIs. For any local processing, data analysis, and interaction with these APIs, the following system configuration was utilized:

- **CPU:** AMD EPYC 9354 32-Core Processor (1 Socket, 32 Cores, 64 Threads).

- **GPU:** NVIDIA RTX 6000 Ada Generation.

    - GPU Memory: 49140 MiB
    - NVIDIA Driver Version: 535.230.02
    - CUDA Version: 12.2

- **System Memory (RAM):** 125 GiB.

- **Operating System Environment:** Linux-based (x86_64 architecture).

This configuration provided a stable and capable environment for managing the data, running evaluation scripts, and interfacing with the LLM APIs. The specific computational requirements for querying each LLM API varied depending on the model provider and were not a limiting factor in this local setup.

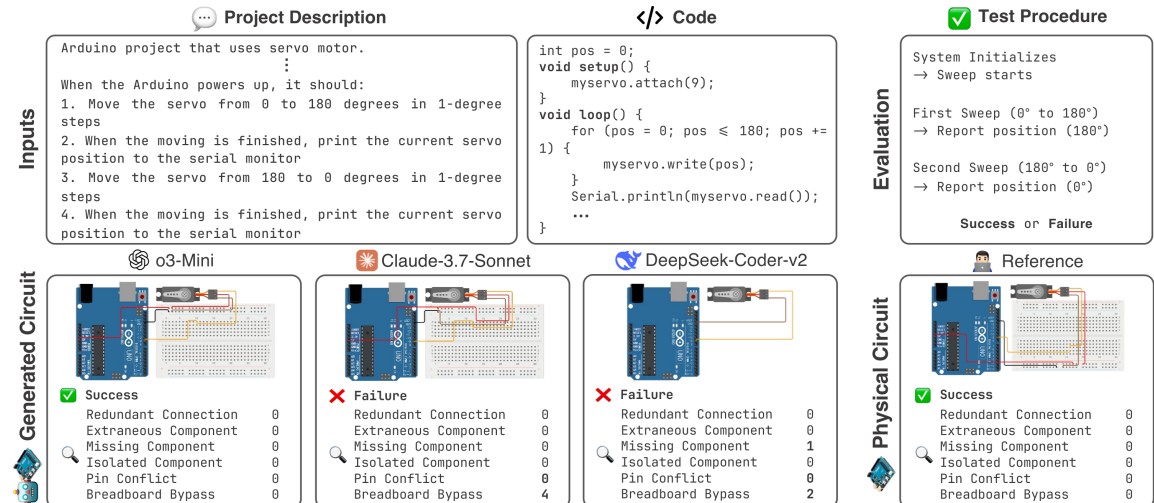

Figure 4: Qualitative examples of LLM-generated physical circuits for a servo motor basic project (level 1), illustrating successful and failed attempts with corresponding error analyses.

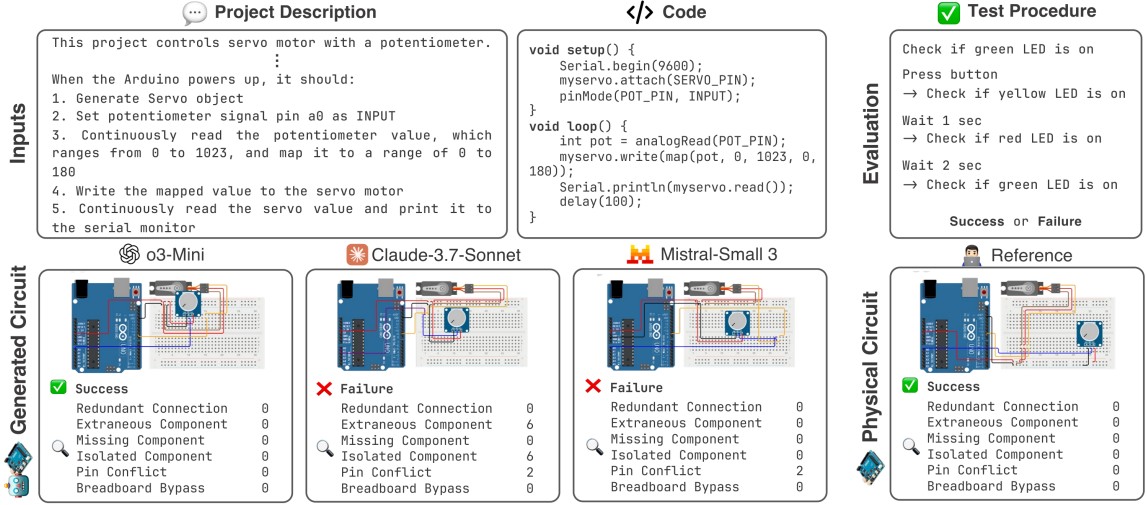

Figure 5: Qualitative examples of LLM-generated physical circuits for a potentiometer servo motor project (level 3), illustrating successful and failed attempts with corresponding error analyses.

## I MORE QUALITATIVE RESULTS

This section presents additional qualitative results to provide a more nuanced understanding of LLM performance in physical circuit generation. The figures below showcase visual examples from specific PCEVAL projects, illustrating both successful circuit implementations and common failure modes encountered by the LLMs. These examples serve to complement the quantitative evaluations by offering concrete instances of the models' outputs and the challenges they face.

## J  Visualizing the Impact of Project Complexity Metrics on LLM Performance

We present a visual analysis of the relationship between project complexity metrics and LLM performance across four physical computing tasks (Figure 6). The visualizations plot LLM success rates (0.0–1.0) against two complexity metrics, number of connections and lines of code. We visualized average performance of top performer LLMs (o3-mini, GPT-4o, Claude 3.7 Sonnet, Gemini-2.0-Flash, Mistral-small 3). Each project appears as a data point color-coded by complexity level (1–4), with linear regression lines (in red) indicating performance trends.

### J.1  Circuit Generation Tasks

Figures 6a and 6b present results for Logical Circuit Generation ($D, C \rightarrow L$) and Physical Circuit Generation ($D, C \rightarrow P$), respectively. Both tasks exhibit negative correlations between success rates and complexity metrics, with connection count demonstrating a stronger negative influence than code length. This suggests that structural complexity of the output circuit presents a greater challenge to LLMs than interpreting input code specifications. The effect is particularly pronounced in physical circuit generation, where models must additionally account for breadboard layouts and physical wiring constraints.

### J.2  Code Generation Tasks

For Code Generation from Logical Circuit ($D, L \rightarrow C$) and Code Generation from Physical Circuit ($D, P \rightarrow C$) (Figures 6c and 6d), the number of code lines correlates with a steeper decline in success rates compared to circuit complexity metrics. This pattern indicates that generating longer, more complex code poses a greater challenge to LLMs than interpreting circuit designs, regardless of whether the input specifications are logical or physical in nature.

### J.3  Summary

Our analysis reveals that LLMs exhibit task-specific sensitivities to different dimensions of complexity. The critical limiting factor appears to be the structural complexity of the artifact being generated rather than the complexity of the artifact being interpreted.

## K  Visualizing the Impact of Project Level on LLM Performance

Figure 7 visualizes the distribution of success rates across the four PCEVAL tasks, revealing important patterns in how LLM performance varies both within and across complexity levels.

### K.1  Circuit Generation Tasks

**Logical Circuit Generation (Figure 7a):** Level 1 shows the highest and most consistent performance (median $\approx 0.98$, narrow IQR), followed by a sharp drop at Level 2 (median $\approx 0.76$) with increased variance. Interestingly, Levels 3 and 4 show similar median performance ($\approx 0.50$ and $\approx 0.68$ respectively) but with very large IQRs. The unexpected slight recovery at Level 4 suggests certain complex projects may have more standardized patterns that LLMs can recognize.

**Physical Circuit Generation (Figure 7b):** This task exhibits the most dramatic performance collapse across levels. Even at Level 1, the median ($\approx 0.54$) is substantially lower than other tasks. Level 2 shows further decline (median $\approx 0.20$) with high variance, while Levels 3 and 4 demonstrate near-floor

performance (medians $\approx 0.08$ and $\approx 0.04$). The compressed box plots at higher levels indicate consistently poor performance with minimal variance.

## K.2 CODE GENERATION TASKS

**Code Generation from Logical Circuit (Figure 7c):** The box plot shows a clear downward trend in median performance from Level 1 (median $\approx 0.96$) to Level 4 (median $\approx 0.40$). Levels 1 and 2 maintain relatively high medians with compact interquartile ranges (IQRs), while Levels 3 and 4 exhibit increased variance with IQRs spanning approximately 0.70. The presence of both perfect scores (1.0) and complete failures (0.0) at each level suggests binary success/failure patterns rather than gradual degradation.

**Code Generation from Physical Circuit (Figure 7d):** This task demonstrates similar degradation patterns but with slightly lower overall performance. The Level 1 median ($\approx 0.88$) remains high but shows more variance than its logical circuit counterpart. The progressive decline is more pronounced, with the Level 4 median dropping to $\approx 0.36$. Outliers at 0.0 across all levels indicate that physical circuit interpretation can fail completely even for simpler projects.

# L   ANALYSIS OF OUTLIER PROJECTS

To understand the factors driving performance variability, we examined two notable outlier projects that deviate significantly from their complexity level expectations.

## L.1   BAR LED BASIC (LEVEL 1)

The "Bar LED Basic" project (Figure 8) achieves 0% success across all tasks despite its Level 1 classification. While conceptually simple—lighting a bar graph LED—the implementation requires:

- 10 LED connections, each with its own resistor

- Coordinated control of multiple pins

- Understanding of the bar graph LED's internal structure

This multiplicative complexity in connections (20 total for a "single" component) overwhelms LLMs' pattern recognition capabilities, causing complete failure even in logical circuit generation.

## L.2   4 DIGIT 7 SEGMENT DISPLAY (LEVEL 3)

The "4 Digit 7 Segment Display" project (Figure 9) represents Level 3 complexity but achieves near-zero performance across all models. The project compounds multiple challenging aspects:

- 13 pins requiring precise connection mapping

- Multiplexing logic to control four digits with shared segments

- Complex timing requirements for digit switching

- Abstract pattern representation (displaying "1, 2, 3, 4")

The combination of high connectivity, multiplexing concepts, and timing constraints creates a complexity that current LLMs hard to navigate.

## M    USE OF LARGE LANGUAGE MODELS

In accordance with ICLR 2026 policy, we disclose the use of Large Language Models in preparing this manuscript. We used LLMs to assist with grammar checking, sentence restructuring, and language polishing during the revision process. The LLM was primarily used to improve clarity and conciseness of technical descriptions, particularly in the methodology and results sections. All scientific content, experimental design, analysis, and interpretations are the original work of the authors. The LLM served solely as a writing aid and did not contribute to the research conception, or core intellectual contributions of this work.

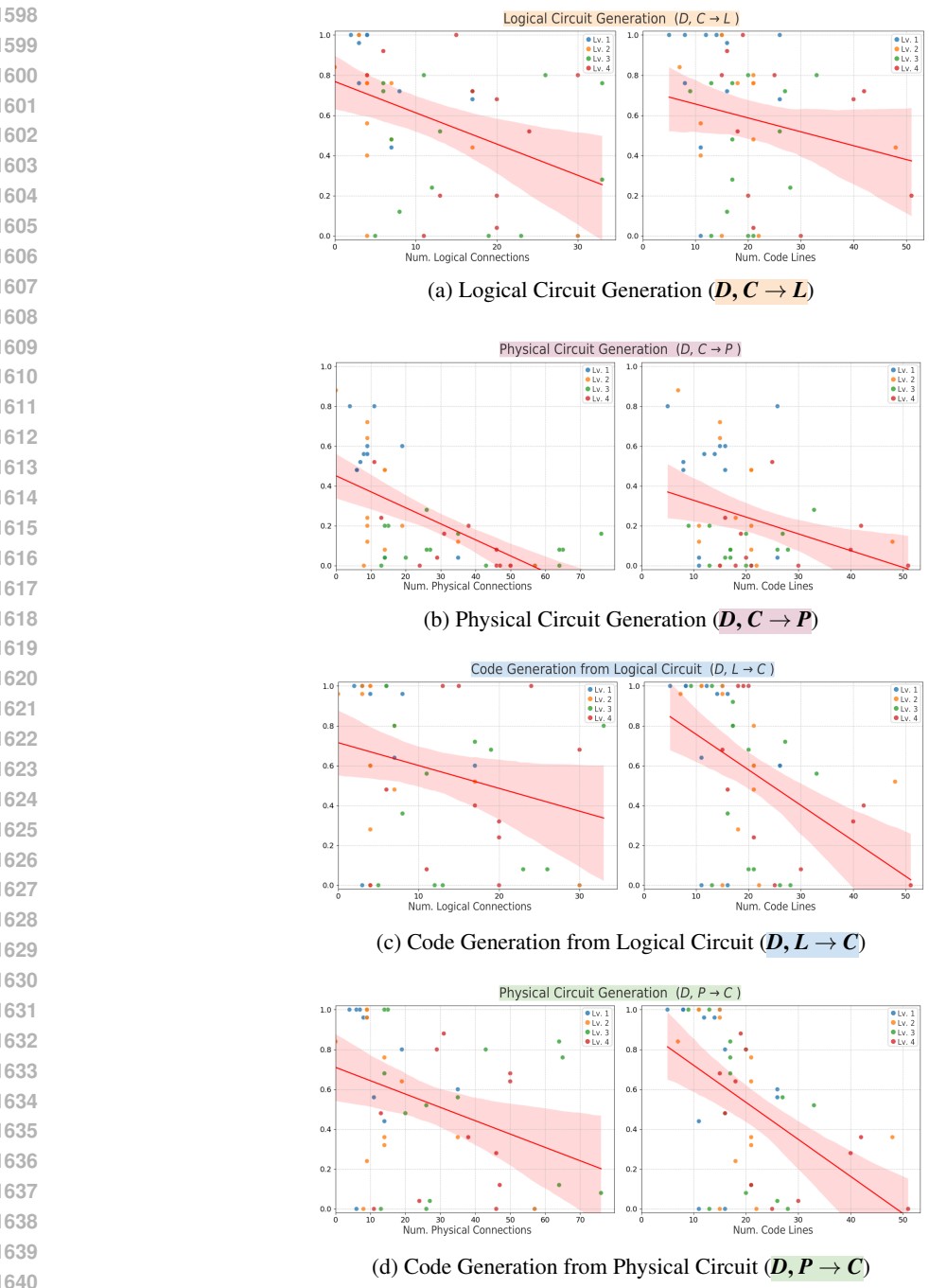

(a) Logical Circuit Generation ($D, C \rightarrow L$)

(b) Physical Circuit Generation ($D, C \rightarrow P$)

(c) Code Generation from Logical Circuit ($D, L \rightarrow C$)

(d) Code Generation from Physical Circuit ($D, P \rightarrow C$)

Figure 6: Test procedure success rates across different tasks and complexity metrics, arranged vertically. Subfigure (a) shows Logical Circuit Generation, (b) Physical Circuit Generation, (c) Code Generation from Logical Circuit, and (d) Code Generation from Physical Circuit. Each subplot details performance against the number of connections and lines of code relevant to the task.

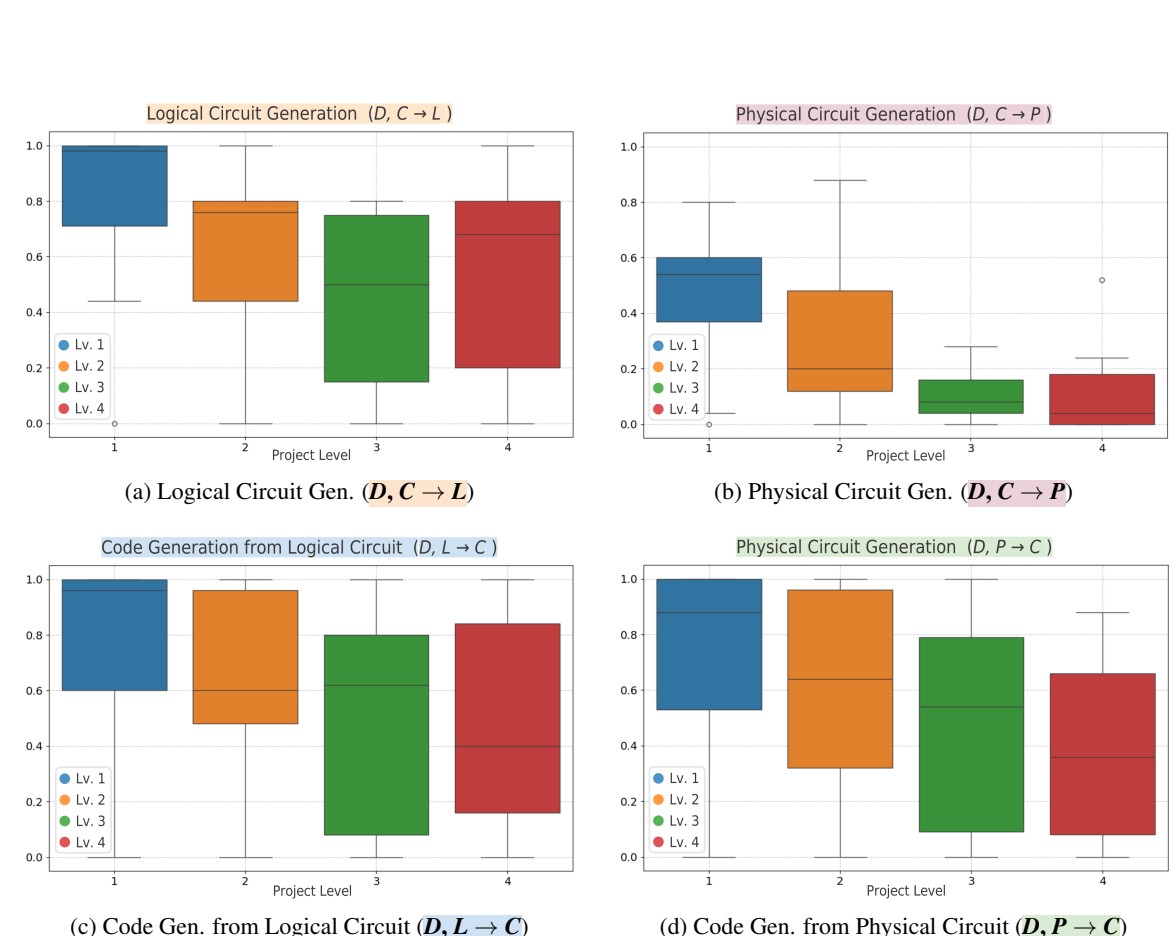

(a) Logical Circuit Gen. ($D, C \rightarrow L$)

(b) Physical Circuit Gen. ($D, C \rightarrow P$)

(c) Code Gen. from Logical Circuit ($D, L \rightarrow C$)

(d) Code Gen. from Physical Circuit ($D, P \rightarrow C$)

Figure 7: Test procedure success rates across different tasks and complexity metrics. Subfigure (a) shows Logical Circuit Generation, (b) Physical Circuit Generation, (c) Code Generation from Logical Circuit, and (d) Code Generation from Physical Circuit. Each subplot details performance against the number of connections and lines of code relevant to the task.

**</> Code**   **Logical Circuit**   **Physical Circuit**

```
int pinList[] = {2, 3, 4, 5, 6, 7, 8, 9, 10, 11};

int threshold = 6;

void setup() {
  for (int i=0; i<10; i++) {
    pinMode(pinList[i], OUTPUT);
  }

  for (int i=0; i<10; i++) {
    if (i < threshold) {
      digitalWrite(pinList[i], HIGH);
    } else {
      digitalWrite(pinList[i], LOW);
    }
  }
}

void loop() {
}
```

Figure 8: Reference code, logical circuit and physical circuit of "Bar LED Basic" project.

**</> Code**   **Physical Circuit**

```
const int segmentPins[8] = {2, 3, 4, 5, 6, 7, 8, 9};
const int digitPins[4] = {10, 11, 12, 13};

const byte digits[10][8] = {
  {1,1,1,1,1,1,0,0},
  {0,1,1,0,0,0,0,0},
  {1,1,0,1,1,0,1,0},
  {1,1,1,1,0,0,1,0},
  {0,1,1,0,0,1,1,0},
  {1,0,1,1,0,1,1,0},
  {1,0,1,1,1,1,1,0},
  {1,1,1,0,0,0,0,0},
  {1,1,1,1,1,1,1,0},
  {1,1,1,1,0,1,1,0}
};

void setup() {
  for (int i = 0; i < 8; i++) pinMode(segmentPins[i], OUTPUT);
  for (int i = 0; i < 4; i++) {
    pinMode(digitPins[i], OUTPUT);
    digitalWrite(digitPins[i], LOW);
  }
}

void loop() {
  int nums[4] = {1, 2, 3, 4};

  for (int i = 0; i < 4; i++) {
    for (int j = 0; j < 8; j++) digitalWrite(segmentPins[j],
digits[nums[i]][j] ? LOW : HIGH);
    digitalWrite(digitPins[i], HIGH);
    delay(800);
    digitalWrite(digitPins[i], LOW);
  }
}
```

**Logical Circuit**

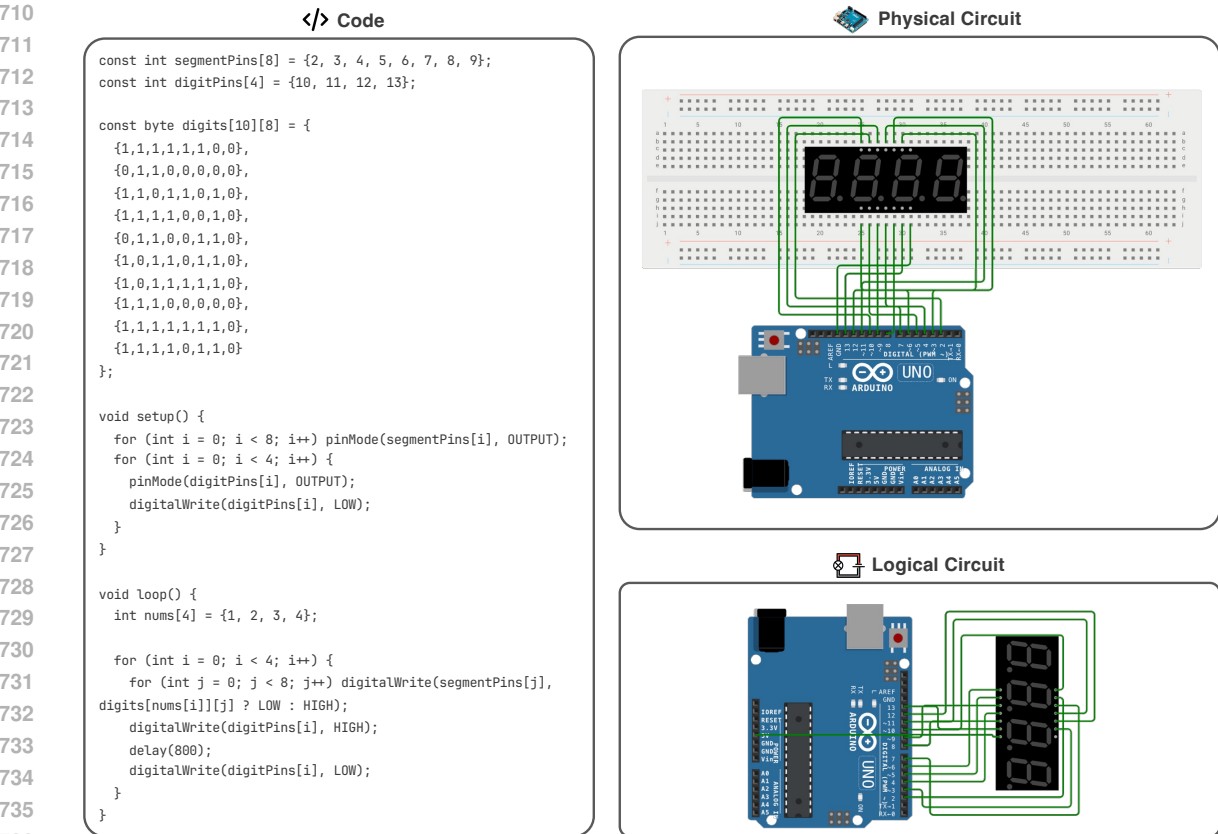

Figure 9: Reference code, logical circuit and physical circuit of "4 Digit 7 Segment Display" project.