# OpenReview forum: "PCEval: A Benchmark for Evaluating Physical Computing Capabilities of Large Language Models"
_ICLR.cc/2026/Conference — Submitted to ICLR 2026_

### Official Review · Reviewer_QYXd · 2025-10-26

**Soundness:** 1
**Presentation:** 2
**Contribution:** 1
**Rating:** 2
**Confidence:** 4

**Summary:**

This work introduces PCEVAL, a benchmark for evaluating large language models (LLMs) in physical computing tasks—where software must interact with and control hardware. Unlike prior benchmarks focusing only on logical or coding abilities, PCEVAL enables fully automated and reproducible evaluation of LLMs in both logical and physical aspects of hardware-related projects, eliminating the need for human assessment.

**Strengths:**

The paper presents a benchmark for evaluating LLM in physical computing tasks.

**Weaknesses:**

1) This work focuses on physical computing but primarily adds components related to breadboard and logic design. The breadboard design tasks are relatively simple and not strongly connected to embedded system design. Moreover, even in cases where breadboard design is necessary, such tasks could likely be automated using traditional algorithms, and it remains unclear why an LLM-based approach is required here.

2) Regarding the logic design aspect, LLM-based methods in this area have already been extensively explored, with numerous existing benchmarks available. The contribution of this work therefore appears to be largely a combination of prior benchmarks. The key distinctions between this benchmark and existing logic design benchmarks should be clearly articulated.

**Questions:**

Please see the weakness.

---

> ### Author Response · Authors · 2025-11-20
> **Author Response to Reviewer QYXd (1/2)**
>
> > **W1. "This work focuses on physical computing but primarily adds components related to breadboard and logic design. The breadboard design tasks are relatively simple and not strongly connected to embedded system design. Moreover, even in cases where breadboard design is necessary, such tasks could likely be automated using traditional algorithms, and it remains unclear why an LLM-based approach is required here."**
>
> We respectfully disagree with the premise that breadboard design tasks are trivial or easily solved by traditional algorithms.
>
> 1. **Task Difficulty**: Empirical Evidence
>
> Our experimental results demonstrate that physical circuit generation is far from simple.
> As shown in Table 3, even state-of-the-art models achieve very low success rates on Physical Circuit Generation (D,C→P): o3-mini at 45.2%, GPT-4o at 26.8%, and most models below 15%.
> This dramatic performance gap compared to Logical Circuit Generation (where the same models achieve 50-66%) indicates that reasoning about physical breadboard constraints presents substantial challenges for current LLMs.
>
> 2. **Domain Clarification**: Physical Computing Education
>
> PCEVAL targets physical computing education rather than Verilog/HDL chip design. We focused on Arduino as it is the most prevalent platform in introductory physical computing curricula [1,2,3]. If the reviewer is referring to a different aspect of embedded system design, we would appreciate clarification.
>
> 3. **Why LLM-based Approach**: End-to-End Generation
>
> Traditional algorithms (e.g., Lee/A* routing [4,5], simulated annealing [6]) require formal netlists and geometric constraints as input—they cannot parse natural language or infer components from descriptions like "create a traffic light system." PCEVAL evaluates this end-to-end generation capability.
>
> Furthermore, automated evaluation of such generation tasks is non-trivial. The most comparable prior work, MICRO25 [7], relied entirely on manual expert evaluation due to these challenges. PCEVAL's contribution includes enabling fully automated validation through simulation
>
> | Aspect | Classical EDA Algorithms                       | PCEVAL Tasks                                    |
> |--------|------------------------------------------------|-------------------------------------------------|
> | Input  | Formal specifications (netlists, constraints)  | Natural language descriptions                   |
> | Task   | Optimize/validate given circuit                | Generate circuit/code from intent               |
>
> [1] Mohammed El-Abd. A review of embedded systems education in the arduino age: Lessons learned and future directions. International Journal of Engineering Pedagogy, 7(2), 2017.
>
> [2] Pedro Antonio García-Tudela and José-Antonio Marín-Marín. Use of arduino in primary education: a systematic review. Education Sciences, 13(2):134, 2023.
>
> [3] Eric Schätz, Lutz Hellmig, and Alke Martens. Analysis of student-problems while working with physical computing devices. In CSEDU (1), pp. 322–329, 2024.
>
> [4] Lee, Chin Yang. "An algorithm for path connections and its applications." IRE transactions on electronic computers 3, 2009.
>
> [5] Hart, Peter E., Nils J. Nilsson, and Bertram Raphael. "A formal basis for the heuristic determination of minimum cost paths." IEEE transactions on Systems Science and Cybernetics 4, no. 2 1968.
>
> [6] Kirkpatrick, Scott, C. Daniel Gelatt Jr, and Mario P. Vecchi. "Optimization by simulated annealing." science 220, no. 4598, 1983.
>
> [7] Peter Jansen. From words to wires: Generating functioning electronic devices from natural language descriptions. In Findings of the Association for Computational Linguistics: EMNLP, 2023.

---

> ### Author Response · Authors · 2025-11-20
> **Author Response to Reviewer QYXd (2/2)**
>
> > **W2. Regarding the logic design aspect, LLM-based methods in this area have already been extensively explored, with numerous existing benchmarks available. The contribution of this work therefore appears to be largely a combination of prior benchmarks. The key distinctions between this benchmark and existing logic design benchmarks should be clearly articulated.**
>
> We respectfully clarify that PCEVAL is fundamentally distinct from existing benchmarks and cannot be replicated by combining prior datasets.
> While LLMs have been tested on code generation [1,2] and logic design [3,4] independently, PCEVAL differs in three key aspects:
>
> 1. different domain—physical computing education rather than general software or chip design;
>
> 2. physical dimension—we uniquely evaluate physical breadboard layout generation with real-world constraints (pin conflicts, breadboard bypasses);
>
> 3. code-circuit coherence—tasks require generating code or circuits that must work together, not in isolation.
>
> As summarized below (recap of Table 1):
> | Benchmark              | Domain          | Code | Logical Circuit | Physical Circuit | Validation      |
> |------------------------|-----------------|------|-----------------|------------------|-----------------|
> | HumanEval/MBPP [1,2]   | General SW      | ✓    | ✗               | ✗                | Unit Testing    |
> | VerilogEval/RTLLM [3,4]| Chip Design     | ✓    | ✓    | ✗                | HDL Simulation  |
> | PCEVAL (Ours)          | Physical Comp.  | ✓    | ✓               | ✓    | HW/SW Simulator |
>
> To our knowledge, no existing benchmark combination can produce PCEVAL's physical circuit generation tasks or code-circuit coherence evaluation. If we have overlooked relevant work, we would appreciate specific references to address in rebuttal.
>
> [1] Mark Chen, Jerry Tworek, Heewoo Jun, Qiming Yuan, Henrique Ponde De Oliveira Pinto, Jared Kaplan, Harri Edwards, Yuri Burda, Nicholas Joseph, Greg Brockman, et al. Evaluating large language models trained on code. arXiv preprint arXiv:2107.03374, 2021.
>
> [2] Jacob Austin, Augustus Odena, Maxwell Nye, Maarten Bosma, Henryk Michalewski, David Dohan, Ellen Jiang, Carrie Cai, Michael Terry, Quoc Le, et al. Program synthesis with large language models. arXiv preprint arXiv:2108.07732, 2021.
>
> [3] Mingjie Liu, Nathaniel Pinckney, Brucek Khailany, and Haoxing Ren. Verilogeval: Evaluating large language models for verilog code generation. In 2023 IEEE/ACM International Conference on Computer Aided Design (ICCAD), pp. 1–8. IEEE, 2023.
>
> [4] Lu, Yao, Shang Liu, Qijun Zhang, and Zhiyao Xie. "Rtllm: An open-source benchmark for design rtl generation with large language model." In 2024 29th Asia and South Pacific Design Automation Conference (ASP-DAC), pp. 722-727. IEEE, 2024.

---

### Official Review · Reviewer_EG1x · 2025-10-27

**Soundness:** 3
**Presentation:** 3
**Contribution:** 2
**Rating:** 4
**Confidence:** 4

**Summary:**

This paper presents a benchmark for evaluation the physical computing capabilities of large language models. Specifically different from prior work, the benchmark also includes new tasks such as logical circuit generation and physical circuit generation (physical circuit layout breadboard implementations). and automatic evaluation protocal. The authors also evaulated on 13 leading models.

**Strengths:**

(1) Novel and well-motivated benchmark addressing previously unexamined domain. Clear task decomposition and fully automated evaluation framework ensuring reproducibility

**Weaknesses:**

(1) Evaluations lack variance measures, such as pass@k metrics etc. LLMs results could be noisy under temperature sampling.

(2) The benchmarks scope is largely Arduino-centric. This raises question about generality and difficulty of the task on real-world example use cases (other than for educational purposes).

(3) It seems the LLM constently make mistakes on problems easily to correct (i.e. physical contraint violation). Would incorporating such feedback in a agentic framework improve the results?

**Questions:**

(1) Some aspects of the automated validation pipeline are insufficiently detailed for replication. Will the benchmark be open-sourced?

(2) Can the authors present more metrics on evaluation such as pass@k etc.?

---

> ### Author Response · Authors · 2025-11-20
> **Author Response to Reviewer EG1x**
>
> > **W1. "Evaluations lack variance measures, such as pass@k metrics etc. LLMs results could be noisy under temperature sampling."**
>
> > **Q2. "Can the authors present more metrics on evaluation such as pass@k etc.?"**
>
> We reported mean success rates over 5 independent trials per task-model combination to account for sampling variance, as we considered consistent output quality important for practical use. However, we agree that pass@k metrics provide additional insight into model capabilities.
> We have computed pass@5 results as shown below, and will incorporate the results into our paper during the discussion period:
>
> | Model                 | D, C → L | D, C → P | D, L  → C | D, P → C | Overall |
> | --------------------- | -------- | -------- | --------- | -------- | ------- |
> | Gemini 2.0 Flash-Lite | 79.2     | 16.7     | 70.8      | 75.0     | 60.4    |
> | Claude 3.7 Sonnet            | 87.5     | 50.0     | 83.3      | 83.3     | 76.0    |
> | GPT-4.1               | 87.5     | 54.2     | 83.3      | 87.5     | 78.1    |
> | Gemma 3            | 70.8     | 8.3      | 41.7      | 33.3     | 38.5    |
> | Phi4                  | 66.7     | 4.2      | 62.5      | 58.3     | 47.9    |
> | Mistral Small     | 75.0     | 13.6     | 62.5      | 50.0     | 50.2    |
>
>
> > **W2. The benchmarks scope is largely Arduino-centric. This raises question about generality and difficulty of the task on real-world example use cases (other than for educational purposes).**
>
> We acknowledge that PCEval's scope is Arduino-centric and focused on educational contexts. This scope was grounded in teacher interviews and actual curriculum practices (Section 3.4), though we recognize it limits applicability to professional scenarios as noted in Section 6 (Limitations).
>
> However, the benchmark's generalizability is supported by two factors: (1) our task decomposition (logical/physical circuit and code generation) and evaluation metrics are platform-independent by design, and (2) we empirically validated portability by testing 24 projects on ESP32 (Appendix G), which showed similar performance trends (Table 10) without any modification to tasks or metrics.
>
> Our future work focuses on both depth (more advanced scenarios beyond educational contexts) and breadth (additional platforms), while maintaining the automated validation pipeline central to PCEval's contribution.
>
> > **W3. "It seems the LLM constently make mistakes on problems easily to correct (i.e. physical contraint violation). Would incorporating such feedback in a agentic framework improve the results?"**
>
> We explored this direction through our self-improvement experiments (Section 4.3), where models received structured failure logs (pin conflicts, missing components, etc.) and iteratively refined their outputs. This approach yielded consistent improvements (e.g., o3-mini: 45.2% → 58.0% on D, C → P).
> We also tested an extended agentic framework with a dedicated Advisor LLM that processes error logs before providing feedback for D, C → P task:
>
> | Model             | Basic | Self-Improvement | Agentic Advisor |
> |------|-------|-----------|---------|
> | o3-mini           | 45.2  | 58.0      | 50.8    |
> | Mistral-Small 24B | 13.6  | 17.6      | 18.4    |
>
> Results suggest that reasoning models like o3-mini benefit more from direct self-improvement, while weaker models may benefit from additional advisory components.
>
> > **Q1. "Some aspects of the automated validation pipeline are insufficiently detailed for replication. Will the benchmark be open-sourced?"**
>
> We apologize if aspects of the validation pipeline were unclear. The benchmark, including the complete evaluation code, dataset, and automated validation scripts, has been open-sourced (anonymous link per blind review guidelines: [Anonymous Link for PCEval-Dataset](https://github.com/Null99-Dog/PCEval-Dataset).
> Due to space constraints, we had to condense the description, which may have resulted in insufficient detail. We will provide more comprehensive documentation during the discussion period. If there are specific aspects that caused confusion, we would be happy to clarify them.

---

### Official Review · Reviewer_b5X5 · 2025-10-28

**Soundness:** 4
**Presentation:** 3
**Contribution:** 4
**Rating:** 8
**Confidence:** 2

**Summary:**

The authors propose PCEval, the first reproducible and verifiable assessment suite to test LLMs in the context of designing physical computing for educational purpose. Empirical evidence suggests that large language models are still insufficient in relevant tasks. It means that we have to use LLMs with caution in physical computing, and more attention should be devoted to this area.

**Strengths:**

1. The investigated question is novel and interesting: how good LLMs are in physical computing for educational purposes. It is well motivated. With the development of LLMs in many fundamental tasks such as reasoning, it is important to understand how useful they are in real-life tasks.

2. The study design is carefully constructed. It starts by interviewing multiple CS educators about what problems are critical in their context, making sure that the investigated problems are relevant to real-world deployment.

3. The proposed framework is scalable and verifiable. This is important in the future impact of this work in relevant and broader areas.

4. The presentation is very clear and easy to follow and understand.

**Weaknesses:**

Mitigation methods can be more thoroughly discussed. For instance, why CoT works on some models but not others? Is there any method to improve model performance instead of simply prompting?

**Questions:**

Please refer to weaknesses above.

---

> ### Author Response · Authors · 2025-11-20
> **Author Response to Reviewer b5X5**
>
> > **W1. "Mitigation methods can be more thoroughly discussed. For instance, why CoT works on some models but not others? Is there any method to improve model performance instead of simply prompting?"**
>
> We applied self-improvement and CoT as they are widely-used approaches for improving LLM performance on complex tasks [1,2].
>
> Regarding why CoT works on some models but not others: we hypothesize this relates to models' varying abilities to leverage intermediate reasoning steps. Models with stronger reasoning capabilities (e.g., o3-mini) may already perform implicit decomposition, making explicit CoT less beneficial, while others benefit more from the structured intermediate representation.
>
> During the rebuttal period, we also evaluated an extended agentic framework for the D, C → P task, incorporating a dedicated Advisor LLM that processes error logs before generating feedback.
>
> | Model             | Basic | Self-Improvement | Agentic Advisor |
> |-------------------|-------|-----------|---------|
> | o3-mini           | 45.2  | 58.0      | 50.8    |
> | Mistral-Small 24B | 13.6  | 17.6      | 18.4    |
>
> Results suggest that reasoning models like o3-mini benefit more from direct self-improvement, while weaker models may benefit from additional advisory components.
> Beyond prompting, training-based approaches would be promising but would require specialized model architectures designed to capture hardware-specific constraints (breadboard topology, pin conflicts, code-circuit coherence). PCEval provides the structured, automatically-validatable foundation to support such future research.
>
> [1] Deng, Shijian, Kai Wang, Tianyu Yang, Harsh Singh, and Yapeng Tian. "Self-Improvement in Multimodal Large Language Models: A Survey." In Findings of the Association for Computational Linguistics: EMNLP, 2025.
>
> [2] Wei, Jason, Xuezhi Wang, Dale Schuurmans, Maarten Bosma, Fei Xia, Ed Chi, Quoc V. Le, and Denny Zhou. "Chain-of-thought prompting elicits reasoning in large language models." Advances in neural information processing systems, 2022.

---

### Official Review · Reviewer_XjH7 · 2025-11-01

**Soundness:** 2
**Presentation:** 2
**Contribution:** 3
**Rating:** 8
**Confidence:** 2

**Summary:**

This paper introduces PCEval, the first benchmark for evaluating the capabilities of large language models in physical computing-that is, reasoning and generating both logical and physical circuits, as well as the code that operates them.
The benchmark breaks down physical-computing reasoning into four tasks:
1. Logical circuit generation (D,C--L), 2. Physical circuit generation (D,C--P), 3. Code generation from logical circuits (D,L--C), 4. Code generation from physical circuits (D,P--C).
PCEval includes 50 Arduino-based projects spanning four complexity levels that can be fully executed in simulation using the Wokwi environment.  Each project includes a test procedure for automated validation, which eliminates subjective expert judgment common in previous works such as MICRO25 or EmbedTask.
Results from 13 leading models show that while LLMs perform reasonably on code generation (60-70% success), they struggle dramatically with physical layout generation (<10% success for most models).

**Strengths:**

- Novel Evaluation Dimension
The paper precisely identifies the missing capability in current LLM evaluation: the ability to reason about and execute tasks that require physical computing.

- Reproducible Evaluation Pipeline
The benchmark's use of fully automated simulation (Wokwi) ensures objective, quantitative validation of the generated circuits and code. This contrasts with previous work (e.g., EmbedTask, MICRO-25), which relied on subjective human grading or partial execution tests. The inclusion of physical-level error metrics (pin conflicts, bypass errors, and isolated components) provides an unusual level of granularity and makes the results understandable to engineering reseachers.

- Comprehensive Model Coverage and Systematic Analysis
The evaluation covers 13 prominent LLMs, including both closed-source (GPT-4o, o3-mini, Gemini-2.5-Pro, Claude 3.7) and open-source (Mistral-Large, Qwen-VL-Max, Llama-3-70B) systems.  Consistent prompting and multi-trial runs produce a reliable cross-model comparison. The analysis goes beyond raw accuracy to reveal failure modes by reasoning stage (for example, layout versus logic consistency), which is extremely useful for model diagnosis.

- Educational and Practical Grounding
Interviews with eight computer science educators guide the benchmark's development, ensuring that the tasks chosen reflect realistic student exercises (sensor integration, actuation control, sequential logic).  This foundation strengthens the benchmark's authenticity and demonstrates its potential educational impact( for example, as a diagnostic or formative assessment tool in engineering courses).

-Exploration of Self-Corrective Techniques
The inclusion of self-improvement and chain-of-thought prompting (resulting in a +10-18% improvement) shows that the benchmark can directly stimulate methodological research rather than serving solely as a static evaluation dataset.

**Weaknesses:**

- Dataset's Scale and Diversity Limitations
With only 50 projects, PCEval's coverage is limited when compared to large-scale code or reasoning benchmarks. The tasks are primarily for introductory Arduino applications (LEDs, sensors, and servos), with less emphasis on other topics such as real-time signal processing.

-Limited Comparison
Lack of comparisons with classical or hybrid design-automation systems (e.g., symbolic circuit solvers, search-based algorithms). This makes it hard for the audience to contextualize LLM weaknesses in relation to domain-specific benchmarks. Such comparisons could help determine whether failures are due to reasoning limitations or a lack of embedded knowledge.

-Evaluation Fairness
Details about prompt standardization (temperature, token limit, and input modality) are not fully specified. Such parameters have a significant impact on model rankings, particularly when comparing multimodal and text-only systems.

**Questions:**

Dataset Balance & Expansion - Given 50 projects, how do you ensure adequate representation of diverse physical computing paradigms (sensing vs actuation control)? Are there plans to scale beyond Arduino while preserving automated validation?

Evaluation Fairness and Reproducibility - Were all LLMs prompted with the same temperature, context length, and output format constraints? How sensitive are results to prompt reformulation (particularly in D,P→C tasks)?

Self-improvement and COT Prompts - How was the iterative refinement protocol developed (failure-log feedback schema)? Did any models show overfitting or oscillatory corrections during the multi-turn refinement?

In the focus-group study, were educators asked to rank task readability or error traceability? Could the authors release annotated examples labeled by usability to facilitate future "AI-pedagogy alignment" studies?

---

> ### Author Response · Authors · 2025-11-20
> **Author Response to Reviewer XjH7 (1/3)**
>
> > **W1. "Dataset's Scale and Diversity Limitations With only 50 projects, PCEval's coverage is limited when compared to large-scale code or reasoning benchmarks. The tasks are primarily for introductory Arduino applications (LEDs, sensors, and servos), with less emphasis on other topics such as real-time signal processing."**
>
> > **Q1. "Dataset Balance & Expansion - Given 50 projects, how do you ensure adequate representation of diverse physical computing paradigms (sensing vs actuation control)? Are there plans to scale beyond Arduino while preserving automated validation?"**
>
> Thank you for this observation. PCEval's scale and scope were determined based on our educational motivation and grounded in actual classroom practices. As described in Section 3.4, teacher interviews revealed that educators "often cover only two to three projects per semester," and our 50 projects were designed to "provide sufficient breadth and depth to cover the scope of a full semester course." The focus on introductory Arduino applications is based on middle/high school STEM curricula.
>
> We acknowledge this limitation in Section 6, noting that PCEval "prioritizes educational contexts, focusing on introductory-level projects common in STEM classrooms" and that this scope "excludes professional physical computing scenarios involving advanced constraints (e.g., power optimization)."
>
> For platform scalability beyond Arduino, we have already conducted cross-platform validation. As detailed in Appendix G, we tested 24 projects on ESP32, and results showed clearly similar trends (Table 10), confirming that our automated evaluation framework transfers effectively without modification.
>
> Our future work focuses on both depth (more advanced scenarios) and breadth (additional platforms), while maintaining the automated validation pipeline central to PCEval's contribution.
>
> > **W2. "Limited Comparison Lack of comparisons with classical or hybrid design-automation systems (e.g., symbolic circuit solvers, search-based algorithms). This makes it hard for the audience to contextualize LLM weaknesses in relation to domain-specific benchmarks. Such comparisons could help determine whether failures are due to reasoning limitations or a lack of embedded knowledge."**
>
> We appreciate the suggestion to contextualize our results against classical systems. However, a direct performance comparison is methodologically infeasible due to fundamental differences in input modality and abstraction level. We clarify these distinctions below with references to established EDA (Electronic Design Automation) literature.
>
> 1. **Input Modality Mismatch** (Natural Language vs. Formal Netlist)
>
> Classical symbolic solvers and search-based algorithms, such as ESPRESSO for logic minimization or Simulated Annealing for placement, are deterministic optimization engines [1,2]. They strictly require formal, structured inputs (e.g., truth tables, netlists, or geometric constraints).
> In contrast PCEVAL starts from ambiguous natural language (e.g., "create a traffic light system").
> Since classical algorithms possess no capability to parse intent or infer necessary components (e.g., "traffic light" requires 3 LEDs and resistors), they cannot perform the end-to-end generation tasks without a human.
>
> 2. **Optimization vs. Generation**
>
> Traditional tools like Lee/A* routing focus on optimization within a closed search space [3,4]. In contrast, PCEVAL evaluates generation in an open semantic space. There is currently no standard symbolic solver available to serve as a baseline.
>
> | Aspect      | Classical EDA Algorithms                                              | PCEVAL Tasks                                    |
> |---|---|---|
> | Input       | Formal specifications (truth tables, netlists) | Natural language descriptions + partial information |
> | Task        | Optimize/validate given circuit                                       | Generate circuit/code                           |
>
> 3. **Clarifying Failure**
>
> The reviewer asked whether failures stem from "reasoning limitations or a lack of embedded knowledge."
> Our results suggest reasoning limitations are more prominent.
>
> As shown in paper Table 3, models achieve reasonable success on Logical Circuit Generation, indicating they possess relevant domain knowledge. However, performance drops dramatically on Physical Circuit Generation.
> This disparity suggests LLMs have embedded knowledge but struggle to reason about physical constraints.
>
> [1] Brayton et al. Logic minimization algorithms for VLSI synthesis. Vol. 2. Springer Science & Business Media, 1984.
>
> [2] Kirkpatrick et al. "Optimization by simulated annealing." science 220, no. 4598, 1983.
>
> [3] Lee, Chin Yang. "An algorithm for path connections and its applications." IRE transactions on electronic computers 3, 2009.
>
> [4] Hart et al. "A formal basis for the heuristic determination of minimum cost paths." IEEE transactions on Systems Science and Cybernetics 4, no. 2 1968.

---

> ### Author Response · Authors · 2025-11-20
> **Author Response to Reviewer XjH7 (2/3)**
>
> > **W3. "Details about prompt standardization (temperature, token limit, and input modality) are not fully specified. Such parameters have a significant impact on model rankings, particularly when comparing multimodal and text-only systems."**
>
> > **Q2. "Evaluation Fairness and Reproducibility - Were all LLMs prompted with the same temperature, context length, and output format constraints? How sensitive are results to prompt reformulation (particularly in D,P→C tasks)?"**
>
> We appreciate this constructive feedback. We agree that temperature, token limits, and input modality details are crucial for reproducibility. We will add a dedicated appendix on hyperparameter settings and prompt sensitivity analysis to the revised paper.
>
> **Input Modality**: All inputs are structured text, and the complete prompts are provided in Appendix F.
> **Token Limit**: We set a token limit of 4,096, which was sufficient for all tasks given their complexity.
> **Temperature**: Due to the large search space, we used each model’s default temperature in our main experiments. Following your suggestion, we conducted additional experiments on two models across different temperature settings.
>
> **(1) Temperature Sensitivity Analysis**
>
> | Temperature | Model             | Overall | D,L→C | D,P→C | D,C→L | D,C→P |
> |-------------|-------------------|---------|-------|-------|-------|-------|
> | 0.1         | Phi 4             | 35.5    | 48.8  | 54.4  | 36.4  | 2.4   |
> |             | Mistral-Small | 41.9    | 52.0  | 49.6  | 50.0  | 16.0  |
> | 0.2         | Phi 4             | 34.8    | 51.7  | 42.5  | 44.2  | 0.8   |
> |             | Mistral-Small | 40.1    | 56.7  | 40.8  | 57.5  | 13.3  |
> | 0.3         | Phi 4             | 36.0    | 50.4  | 48.8  | 42.0  | 2.8   |
> |             | Mistral-Small | 40.7    | 51.2  | 51.6  | 49.6  | 10.4  |
>
> Recent studies suggest lower temperatures favor logical tasks like coding [1,2]. Our results show Mistral-Small follows this trend, while Phi-4 does not.
> Importantly, the performance variance across temperatures is relatively small (±1.2%), and both the relative ranking between models and the difficulty ordering across task categories (with Physical Hardware being the hardest) remain consistent.
>
>
> **(2) Effect of Providing Breadboard Electrical Topology**
>
> To evaluate prompt sensitivity, we tested whether removing the explicit breadboard electrical connectivity information from our original prompts affects performance on tasks requiring physical reasoning (D,C→P and D,P→C).
>
> | Model | Avg. | D,P→C | D,C→P |
> |-------|------|--------|--------|
> | gemini-2.0-flash | 37.8 → 50.4 | 54.4 → 54.4 | 21.2 → 44.8 |
> | gpt-4o-mini       | 27.4 → 40.2 | 53.6 → 50.8 | 1.2 → 29.6 |
>
> This result confirms that explicitly specifying breadboard electrical connectivity rules in the prompt is critical for physical circuit generation tasks.
>
> We will extend these experiments to additional models during the discussion period and include comprehensive results in the revised paper.
>
>
> [1] Renze, Matthew. "The effect of sampling temperature on problem solving in large language models." In Findings of the association for computational linguistics: EMNLP 2024, pp. 7346-7356. 2024.
>
> [2] Li, Lujun, Lama Sleem, Niccolò Gentile, Geoffrey Nichil, and Radu State. “Exploring the Impact of Temperature on Large Language Models: Hot or Cold?” Procedia Computer Science 264, 2025.

---

> ### Author Response · Authors · 2025-11-20
> **Author Response to Reviewer XjH7 (3/3)**
>
> > **Q3. "Self-improvement and COT Prompts - How was the iterative refinement protocol developed (failure-log feedback schema)? Did any models show overfitting or oscillatory corrections during the multi-turn refinement?"**
>
> As described in Section 4.3, both protocols were motivated by specific observations: the self-improvement approach addressed simple but recurring errors (missing connections, incorrect pin numbers), while CoT prompting aimed to bridge logical design and physical implementation by having models first generate intermediate logical circuit representations—mirroring how humans reason from abstract connections to concrete breadboard placement.
>
> **Regarding oscillatory corrections**: the table below shows the distribution of eventual successes by iteration (Iter 1 = initial attempt without refinement; Iter 2+ = after receiving failure feedback).
>
> | Model         | Iter 1       | Iter 2     | Iter 3     | Iter 4     | Iter 5    |
> |---------------|--------------|------------|------------|------------|-----------|
> | o3-mini       | 654 (85.5%)  | 76 (9.9%)  | 20 (2.6%)  | 11 (1.4%)  | 4 (0.5%)  |
> | Mistral-Small | 370 (85.5%)  | 37 (8.6%)  | 17 (3.9%)  | 5 (1.2%)   | 4 (0.9%)  |
>
> This concentration of successes in early iterations suggests models make genuine corrections rather than oscillating between errors—if such behavior were prevalent, we would expect successes more distributed across later turns.
>
> > **Q4. "In the focus-group study, were educators asked to rank task readability or error traceability? Could the authors release annotated examples labeled by usability to facilitate future "AI-pedagogy alignment" studies?"**
>
> As described in Appendix B.3, pre-service teachers were given reference physical circuits alongside LLM-generated circuits and asked to rate **Readability**, **Correctness**, and **Educational Value** on a 1-5 scale across different complexity levels.
> We also included open-ended questions about primary obstacles for educational use and areas requiring improvement.
>
> We appreciate the insightful suggestion regarding AI-pedagogy alignment research. The detailed survey instruments from our focus group study have been released alongside the dataset (anonymous link per blind review guidelines: [Anonymous Link for PCEval-Dataset](https://github.com/Null99-Dog/PCEval-Dataset)). We believe these materials could facilitate future research on aligning AI outputs with pedagogical requirements.

---

### Meta-Review · Area_Chair_mR2e · 2026-01-07

**Summary:**

This paper introduces PCEval, the first fully-automated benchmark that evaluates whether large language models can jointly generate logical circuits, physical breadboard layouts, and executable Arduino code for educational physical-computing projects. However, the submission raised significant concerns. Reviewers uniformly criticized the limited scope (50 introductory Arduino tasks), the absence of baseline comparisons against classical EDA or search-based layout tools, and the lack of variance measures in the main tables. Although the rebuttal clarified some aspects such as such as cross-platform validation, core concerns about the narrow application scope of the proposed benchmark remained incompletely resolved, leading to the inadequate demonstration of the benchmark’s value in the domain. Based on the above considerations, I recommend rejection, while encouraging the authors to further develop this promising direction.

**Reviewer Concerns:**

Addressed
  - Reviewer XjH7: prompt standardization, temperature sensitivity, variance, self-improvement protocol, dataset scale rationale, comparison with classical EDA.
  - Reviewer b5X5: deeper discussion of mitigation methods and why CoT helps some models; new agentic-advisor results added.
  - Reviewer EG1x: added metrics, confirmed cross-platform transfer, clarified agentic feedback experiments.

Still outstanding
  - Reviewer QYXd: continues to view bread-board tasks as “too simple” and the benchmark as a re-packaging of prior logic-design sets

**Reviewer Scores:**

- XjH7: keep positive
- b5X5: keep positive
- EG1x: 4 → 6
- QYXd: keep negative

---

### Decision · Program_Chairs · 2026-01-26

Reject